# Rhinovirus infection of airway epithelial cells uncovers the non-ciliated subset as a likely driver of genetic risk to childhood-onset asthma

## Graphical abstract

## Authors

Sarah Djeddi, Daniela Fernandez-Salinas, George X. Huang, ..., James E. Gern, Nora A. Barrett, Maria Gutierrez-Arcelus

## Correspondence

mgutierr@broadinstitute.org

## In brief

Djeddi, Fernandez-Salinas, and colleagues gained new mechanistic insights into asthma by integrating genome-wide association study data with transcriptomic datasets of airway epithelial cells subject to different stimuli, including rhinovirus, influenza, SARS-CoV-2, and cytokines. They found that rhinovirus-infected non-ciliated airway epithelial cells are a likely driver of genetic susceptibility to childhood-onset asthma. These findings highlight the importance of gene-by-environment interactions in asthma development.

## Highlights

- Integration of airway epithelial cell transcriptomics with GWAS for asthma traits

- Genes induced by rhinovirus are enriched in childhood-onset asthma risk loci

- Non-ciliated airway epithelial cells are the subset driving this enrichment

- Not all respiratory viruses induce this relevant airway epithelial cell state for asthma

 Djeddi et al., 2024, Cell Genomics 4, 100636
September 11, 2024 © 2024 The Author(s). Published by Elsevier Inc.

CellPress

## Article

# Rhinovirus infection of airway epithelial cells uncovers the non-ciliated subset as a likely driver of genetic risk to childhood-onset asthma

Sarah Djeddi,[1,2,3,12] Daniela Fernandez-Salinas,[1,2,3,4,12] George X. Huang,[5,6] Vitor R.C. Aguiar,[1,2,3] Chitrasen Mohanty,[10] Christina Kendziorski,[10] Steven Gazal,[7,8] Joshua A. Boyce,[5,6] Carole Ober,[9] James E. Gern,[10,11] Nora A. Barrett,[5,6] and Maria Gutierrez-Arcelus[1,2,3,13,*]

[1]Division of Immunology, Boston Children's Hospital, Boston, MA 02115, USA
[2]Department of Pediatrics, Harvard Medical School, Boston, MA 02115, USA
[3]Broad Institute of MIT and Harvard, Cambridge, MA 02142, USA
[4]Licenciatura en Ciencias Genómicas, Instituto de Biotecnología, Universidad Nacional Autónoma de México (UNAM), Cuernavaca, Morelos 62210, México
[5]Department of Medicine, Harvard Medical School, Boston, MA 02115, USA
[6]Jeff and Penny Vinik Center for Allergic Disease Research, Division of Rheumatology, Immunology, and Allergy, Brigham and Women's Hospital, Boston, MA 02115, USA
[7]Department of Quantitative and Computational Biology, University of Southern California, Los Angeles, CA 90007, USA
[8]Norris Comprehensive Cancer Center, Keck School of Medicine, University of Southern California, Los Angeles, CA 90007, USA
[9]Department of Human Genetics, University of Chicago, Chicago, IL 60637, USA
[10]Department of Biostatistics and Medical Informatics, University of Wisconsin-Madison, Madison, WI 53726, USA
[11]Departments of Pediatrics and Medicine, University of Wisconsin School of Medicine and Public Health, Madison, WI 53726, USA
[12]These authors contributed equally
[13]Lead contact
*Correspondence: mgutierr@broadinstitute.org

## SUMMARY

Asthma is a complex disease caused by genetic and environmental factors. Studies show that wheezing during rhinovirus infection correlates with childhood asthma development. Over 150 non-coding risk variants for asthma have been identified, many affecting gene regulation in T cells, but the effects of most risk variants remain unknown. We hypothesized that airway epithelial cells could also mediate genetic susceptibility to asthma given they are the first line of defense against respiratory viruses and allergens. We integrated genetic data with transcriptomics of airway epithelial cells subject to different stimuli. We demonstrate that rhinovirus infection significantly upregulates childhood-onset asthma-associated genes, particularly in non-ciliated cells. This enrichment is also observed with influenza infection but not with severe acute respiratory syndrome coronavirus 2 (SARS-CoV-2) or cytokine activation. Overall, our results suggest that rhinovirus infection is an environmental factor that interacts with genetic risk factors through non-ciliated airway epithelial cells to drive childhood-onset asthma.

## INTRODUCTION

Asthma is a complex and heterogeneous disease that affects 300 million children and adults worldwide and represents a significant burden to healthcare ($82 billion for the US in 2013).[1] Although asthma can develop at any point in a person's life, it commonly begins in childhood. Longitudinal epidemiologic studies have demonstrated many risk factors for childhood-onset asthma. These include familial risk factors such as maternal and paternal asthma; maternal smoking and stress; perinatal risk factors such as preterm birth, low birth weight, and cesarian-section delivery; postnatal exposures including smoke, pollution, indoor allergens, and reduced microbiome diversity; and infections with respiratory syncytial virus and rhino-

virus (RV).[2,3] In the case of RV, wheezing with viral infection is a risk factor for developing asthma later in childhood.[4-9] These observations have led to two hypotheses: (1) RV infection could be causal in asthma development or (2) RV-induced wheeze is a biomarker that identifies children at increased risk for asthma development.

Asthma is significantly influenced by genetics, with heritability estimates starting at 35% and going up to 95%.[10] Genome-wide association studies (GWASs) have discovered more than 150 risk loci for asthma.[11-13] SNP-based heritability estimates in childhood-onset asthma (COA) are two to three times higher than that for adult-onset asthma (AOA).[14,15] Accordingly, the number of discovered risk loci for COA is higher than for AOA.[11,14,15] Furthermore, the genetic correlation estimates

between COA and AOA range from 0.63–0.78, indicating both shared and disease-subset-specific factors.[11,15] One of the most significant loci for COA is located at the 17q21-q12 locus, where *ORMDL3* and *GSDMB* have been nominated as candidate causal genes.[12,13] Notably, the association of 17q21-q12 variants with asthma is restricted to children who wheeze with RV, suggesting important gene-by-environment interactions that are poorly understood.[9]

Similar to other complex diseases, the vast majority of the likely causal risk variants for asthma are non-coding. As a consequence, deciphering the mechanisms through which the risk alleles lead to disease is challenging. Investigators have discovered key cell types mediating genetic susceptibility to complex diseases using a suite of recently developed methods that integrate GWAS data with functional genomics datasets. For example, risk variants for rheumatoid arthritis (RA) are enriched in regulatory elements specific for CD4 T cells, and studies in patients and mice have shown the relevance of these cells in the pathogenesis of this disease.[16–20] Additionally, GWAS integration with transcriptomics revealed that a significant proportion of the risk alleles for Alzheimer's disease (AD) act through the myeloid lineage rather than the brain.[21] AD is now considered an immune-mediated disease.[21] For asthma, T cell-specific regulatory elements and gene expression are enriched in genetic risk loci,[19,20,22,23] with some highlighting particularly T helper type 2 (Th2) cells consistent with the role of type 2 inflammation in asthma pathogenesis.[24–27]

The observations that non-coding risk variants affect gene regulation in cell types relevant to each disease have motivated large-scale transcriptomic studies to identify genetic variants associated with gene expression (expression quantitative trait loci [eQTLs]) and disease risk. However, only 25%–40% of risk variants for immune-mediated diseases co-localize with eQTLs in immune cells.[28–30] For asthma, alleles at only ~47% of loci co-localize with leukocyte expression and/or splicing QTLs.[28] Hence, the regulatory effects of most non-coding risk variants remain unknown. Recent studies have highlighted that these "missing regulatory effects" could be hidden in specific activation or differentiation cell states that have not been systematically ascertained.[31–34]

GWAS enrichment studies have been highly biased toward annotations of blood immune cell types, with reduced resolution when using other tissues relevant to the context of asthma, such as GTEx tissues from postmortem human organs.[14,18,20,22,35] Here, we sought to define whether airway epithelial cell states could be driving genetic susceptibility to asthma and may mediate the observed association of RV-elicited wheeze with COA. We analyzed 10 single-cell and bulk transcriptomic datasets of epithelial cells subject to different activation conditions including infection with human RV. We integrated these datasets with GWAS summary statistics for COA, AOA, unspecified-onset asthma (UOA), and a genetically correlated type 2 inflammation trait: allergy/eczema (Table S1).[36–38] We additionally tested three control traits to assess the specificity of our findings. We used state-of-the-art methods that control for linkage disequilibrium, take advantage of most ascertained genetic variants in the genome, and have been shown to work well for bulk and single-cell datasets.[19,23]

## RESULTS

We applied two methods that use GWAS data to identify relevant cell types for disease. Linkage disequilibrium score regression in specifically expressed genes (LDSC-SEG) identifies heritability enrichment in genomic annotations (such as genes or chromatin marks with specific presence in a particular cell type or cell state).[20] Single-cell disease-relevance score (scDRS) identifies cells, from single-cell RNA sequencing (scRNA-seq) data, that significantly express genes in GWAS loci (weighted according to their strength of association with disease) relative to null sets of control genes in the same dataset.[23] We retrieved GWAS summary statistics from asthma-related traits as described above. Throughout our analyses, we included three complex traits as controls: height as a non-immune control, AD as a trait implicating myeloid cells, and RA as a lymphocyte-driven disease with a strong T cell component.[18,20,22,23,35] For both asthma and RA, activated T cells have been shown to be a relevant cell state[22,35]; however, RA is an autoimmune disease that affects the joints, so a genetic enrichment for RA in gene expression of airway epithelial cell states is not expected.

### T cell involvement in asthma is confirmed by methods integrating cell-type-specific profiles and GWAS data

First, we sought to confirm the effectiveness of the methods to identify T cells as relevant for the genetic susceptibility to asthma, as previously reported in the literature,[20,22,23,35] and to test the methods' resolution to pinpoint the specific T cell states and subsets driving the signal. Applying LDSC-SEG to bulk assay for transposase-accessible chromatin with sequencing (ATAC-seq) data of human peripheral blood leukocyte populations, we confirmed that T cell-specific open-chromatin regions are significantly enriched in heritability for asthma-related traits (Figures S1A–S1D and Table S2).[22] Furthermore, when comparing cell types between their resting and activated states, we confirmed that activation-specific open chromatin in T cells has significant heritability enrichment for all the asthma-related traits (Figure S1E and Table S2). Next, we applied scDRS to scRNA-seq datasets to identify cells with significant overexpression of risk genes identified from GWAS studies (see STAR Methods).[39,40] In sinonasal mucosa tissue from healthy donors and chronic rhinosinusitis patients, we observed that 21%–83% of cells with significant disease-relevant score (10% false discovery rate [FDR]) for asthma-related traits are T cells (Figures S2A–S2F).[39] AOA did not have any cells with significant disease-relevant score at 10% FDR; however, at 20% FDR, 294 out of 405 cells (73%) are T cells. In addition, we found that 67% of cells for COA are myeloid cells (10% FDR). In a dataset of house-dust-mite-activated T cells from asthma and allergic patients, we observed that, among the cells with significant disease-relevant score, most were T effector and Th2 cells (51%–57% and 48%–42% respectively; Figures S3A–S3C).[40] COA did not have any cells with significant disease-relevant score at 10% FDR. Overall, these analyses confirm the validity of the methods used for this study and confirm previous findings showing the relevance of T cells in the genetic susceptibility to asthma-related traits.

**A**

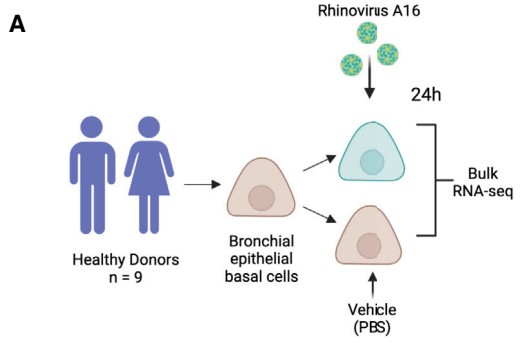

**D**

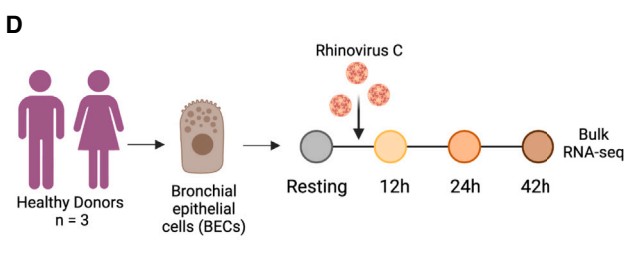

**B**

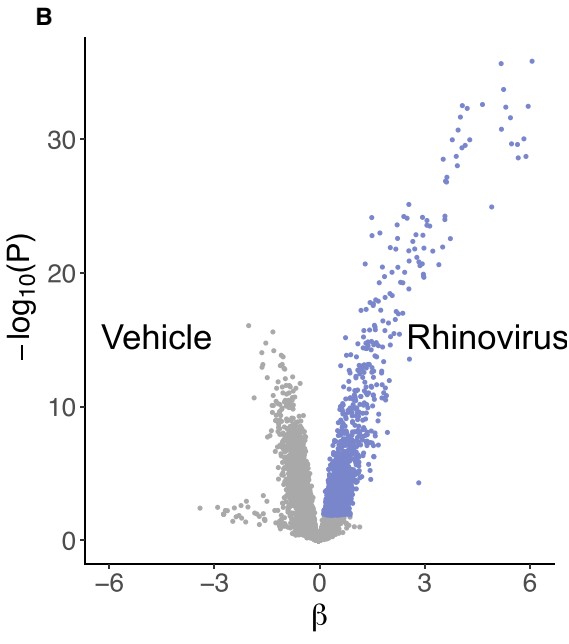

**E**

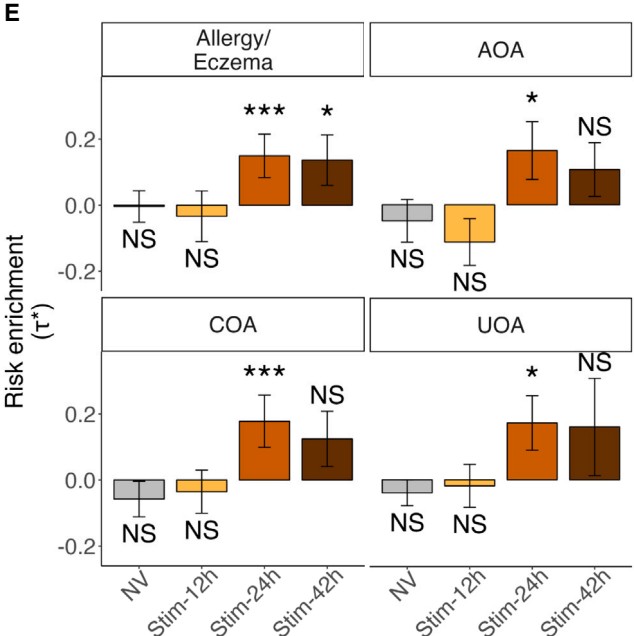

**C**

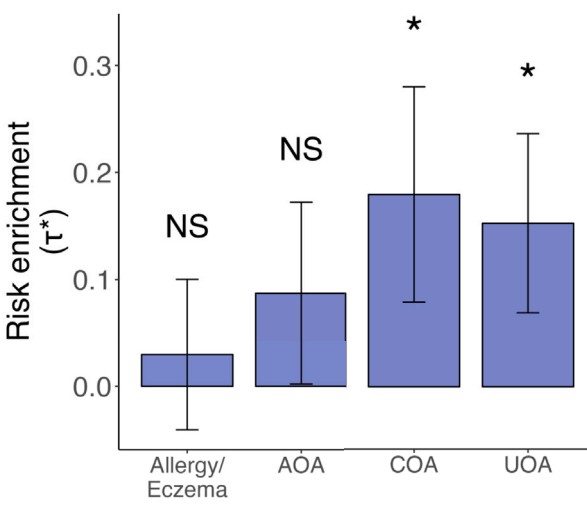

*(legend on next page)*

## Infection with RV-A16 induces upregulation of asthma-associated genes in epithelial cells from healthy donors

To assess the role that RV infection could play in asthma genetic susceptibility at the epithelial cell level, we analyzed publicly available bulk RNA-seq data of basal bronchial epithelial cells (BEC) from healthy donors (n = 9) that were infected *in vitro* with RV species RV-A, subtype 16 (RV-A16) or treated with phosphate buffer saline (PBS) vehicle control (Figure 1A).[41] Using a linear mixed model, we performed differential expression analysis (DEA) testing for RV infection versus PBS vehicle. From the 14,883 tested genes, we selected the top 10% based on t-statistic (as recommended by LDSC-SEG) to select genes that are upregulated with RV infection (Figure 1B and Table S3). We used LDSC-SEG to investigate whether this gene set showed an enrichment for asthma heritability. Our analysis showed significant heritability enrichment in RV-upregulated genes for UOA (p = 0.033) and COA (p = 0.037), with a larger coefficient observed for COA (Figure 1C and Table S2). Moreover, we did not observe any significant enrichment for any of the control traits (AD, RA, height) (Figure S4A and Table S2). The fact that we did not observe significant heritability enrichment in RV-upregulated genes for RA suggests that the signal observed for UOA and COA is not due to a general immune transcriptional response, but rather is a response that is specific for RV infection of epithelial cells.

## Infection with RV-C15 induces upregulation of asthma-associated genes in epithelial cells from healthy donors

Next, we sought to validate these findings in an independent study using a different strain of RV. We reanalyzed a bulk RNA-seq time-course dataset of epithelial cells from three healthy donors where cells were infected with RV-C15, and samples were collected before infection and at 12, 24, and 42 h post-infection (Figure 1D).[42] We performed DEA to identify genes upregulated specifically in each time point (versus all others) and applied LDSC-SEG (Figure S4B and Table S2). We observed significant heritability enrichment for upregulated genes by RV infection specifically at 24 h for all asthma-related traits (p < 0.05; Figure 1E and Table S2). This enrichment was higher for COA and allergy/eczema compared to UOA and AOA (Figure 1E and Table S2). Genes that were specifically expressed at 42 h post infection also had positive enrichment for heritability in all traits, but this was only significant for allergy/eczema (p = 0.03). While the lack of significance at 42 h could be partly due to limited statistical power, we investigated which genes could be involved in the difference observed between 24 and 42 h. We extracted the genes represented in the 24-h-specific annotation but absent in the 42-h-specific annotation and that were at

GWAS risk loci (within 250 kb of the lead variant). We found 27 24-h-exclusive GWAS genes for COA and five for AOA (Tables S4 and S5). Among these is *TSLP*, which encodes for a T2 cytokine and has been associated with both asthma COA and AOA endotypes (Figure S4C).[14,15,43,44] Once again, this enrichment was not present in any of the control traits (Figure S4D and Table S2). Together, these findings suggest that RV-infected epithelial cells represent a cell state that may mediate genetic susceptibility to asthma, with a greater contribution to COA than to AOA.

## RV-induced asthma-associated genes are specifically enriched in non-ciliated epithelial cells

We then asked whether there were specific epithelial cell subsets that may mediate asthma genetic risk after RV infection. To evaluate this, we used scRNA-seq data of three healthy donors, where airway epithelial cell samples were infected with RV-C15 or resting and profiled at 24 h (Figures 2A and S5A–S5C).[42] We performed cell clustering and then annotated the clusters based on epithelial cell markers (Figure S5D). We identified two ciliated cell subsets and five non-ciliated cell subsets: basal, deuterosomal, neuroendocrine, secretory, and transitional (Figures 2B and S5D). We applied scDRS on this dataset to identify cells with significant over-expression of asthma-associated genes. The number of cells with significant disease-relevant scores (10% FDR) varied per trait: 147 cells for UOA, 850 cells for COA, none for AOA, and 147 cells for allergy/eczema (Figure 2C). Notably, COA was the trait with the highest proportion of disease-relevant cells. Among the cells with a significant disease-relevant score, 99% corresponded to the stimulated condition. Furthermore, the disease-relevant cells were strongly over-represented in the non-ciliated cell subsets, representing 96%–99% of the significant cells, while they make up 70% of the whole dataset (Figures 2C and 2D). As expected, we did not observe any significant cells for height, AD, and RA (Figure S5E).

Overall, we identified non-ciliated cells as the main epithelial cell subset with significant upregulation of COA-associated genes after RV infection. However, only ciliated cells are known to be infected by RV-C15, which we confirmed by looking at the expression of the RV-C15 receptor (*CDHR3*) and the presence of the viral sequence itself in the scRNA-seq data (Figure S5F).[45] We therefore hypothesized that ciliated cells may communicate with non-ciliated cells upon RV infection. To investigate some of the possible ligand-receptor mechanisms through which cells may be communicating, we used CellPhoneDB, which uses scRNA-seq data to identify expressed genes encoding ligands and receptors expressed by different cell clusters.[46]

---

**Figure 1. Infection with RV induces upregulation of asthma-associated genes in airway epithelial cells from healthy donors**

(A) Experimental design of the Helling et al.[41] dataset consisting of basal BECs from healthy donors stimulated with PBS or RV-A16.

(B) Volcano plot showing differentially expressed genes after RV infection; genes selected based on t-statistic are colored in purple.

(C) Bar plot showing LDSC-SEG heritability enrichment coefficient (τ*) for each of the asthma-related diseases. Error bars represent τ* ± standard error. *p < 0.05; NS, nonsignificant (p > 0.05).

(D) Experimental design of the Basnet et al.[42] time-course dataset consisting of BECs stimulated with RV-C15.

(E) Bar plot showing LDSC-SEG heritability enrichment coefficient (τ*) for differentially expressed genes at each time point when compared against all others. Error bars represent τ* ± standard error. *p < 0.05, ***Bonferroni-adjusted p < 0.05; NS, nonsignificant (p > 0.05).

AOA, adult-onset asthma; COA, childhood-onset asthma; UOA, unspecified-onset asthma.

See also Figure S4 and Tables S2–S5.

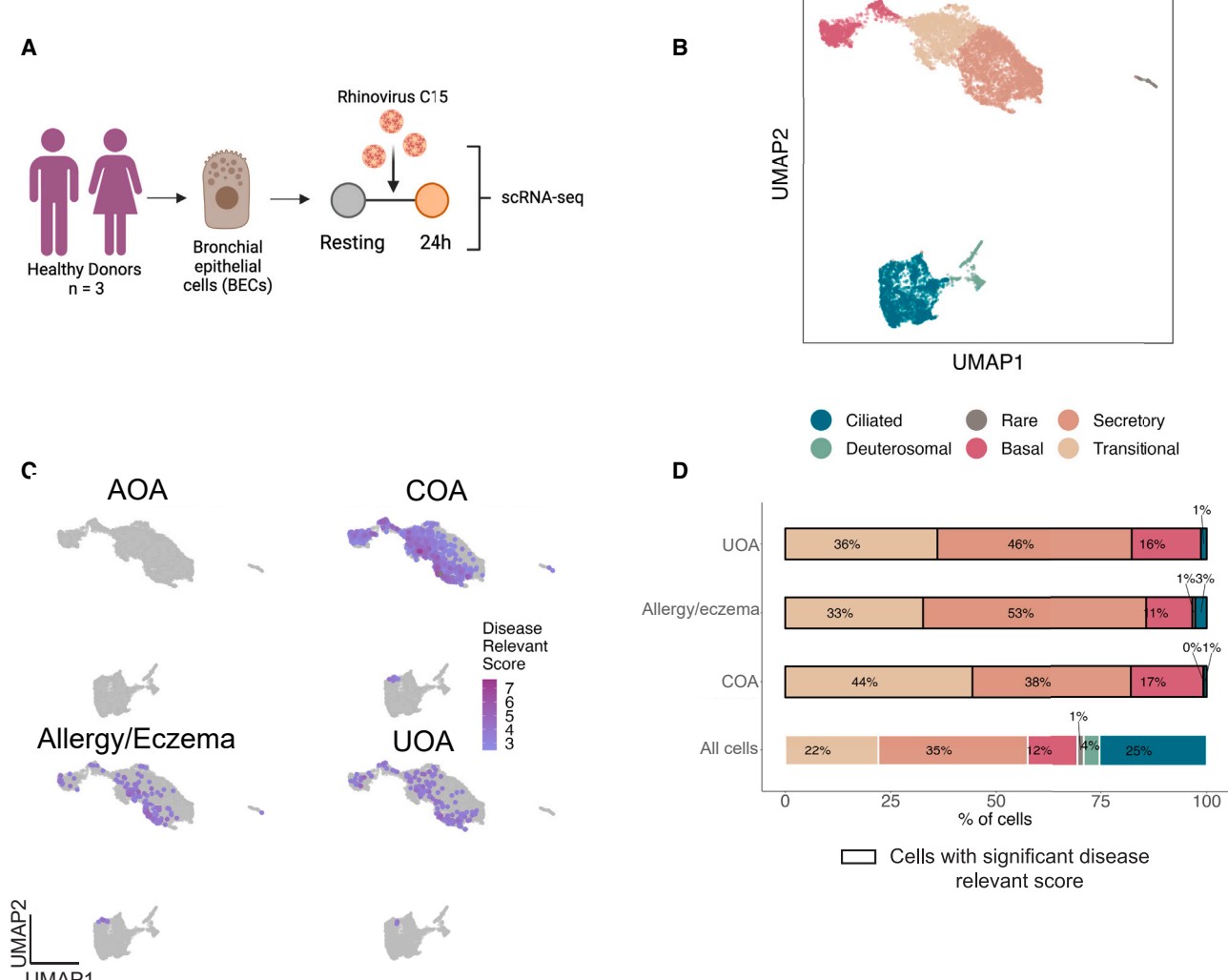

**Figure 2. scRNA-seq uncovers non-ciliated epithelial cells as potential mediators for asthma risk**

(A) Experimental design of the Basnet et al.[42] scRNA-seq dataset of BECs from healthy donors infected with RV-C15.

(B) Uniform manifold approximation and projection (UMAP) visualization of the 10,721 airway epithelial cells colored by cell type.

(C) scDRS results represented on the UMAP for the four asthma-related diseases tested. Cells with significant disease-relevant score at 10% FDR are depicted in purple, with the intensity of the color representing the magnitude of the score. Cells with nonsignificant score are depicted in gray.

(D) Bar plot representing the percentage of each cell type in all cells in the dataset (bottom bar) and in cells with significant scDRS (10% FDR) for COA, allergy/eczema, and UOA. AOA is not represented because no cells were significant.

See also Figures S5 and S6.

Specifically, we looked for ligand-encoding genes expressed in RV-infected ciliated cells and their corresponding receptor-encoding gene expressed in non-infected non-ciliated cells, with the rationale that the receptor should already be expressed in the non-infected state, ready to receive the signal by the ligand produced by ciliated cells upon direct infection by the virus (Figure S6A). Furthermore, we required that the ciliated cell ligand-encoding gene is significantly upregulated upon RV infection (5% FDR, log2 fold change >0.5). We identified 11 candidate pairs of interactors (Figures S6B and S6C). Among these ligand-receptor pairs are TNFSF10-TNFRSF10B. *TNFSF10* encodes for the tumor necrosis factor-related apoptosis-inducing ligand (TRAIL) cytokine, which belongs to the tumor necrosis factor (TNF) ligand family, and it serves as ligand to the receptor encoded by TNFRSF10B (also called DR5, TRAILR2, or CD262), which can induce apoptosis. A study using an RV mouse model and *in vitro* activation of a human airway epithelial cell line suggested that TRAIL promotes RV-induced airway hyperreactivity and inflammation.[47] Another two pairs of ligand-receptor interactors that we identified involve the epidermal growth factor receptor (EGFR), which has been reported to be induced by RV and other respiratory viruses and utilized for suppressing interferon (IFN) signaling and inducing higher viral replication.[48,49]

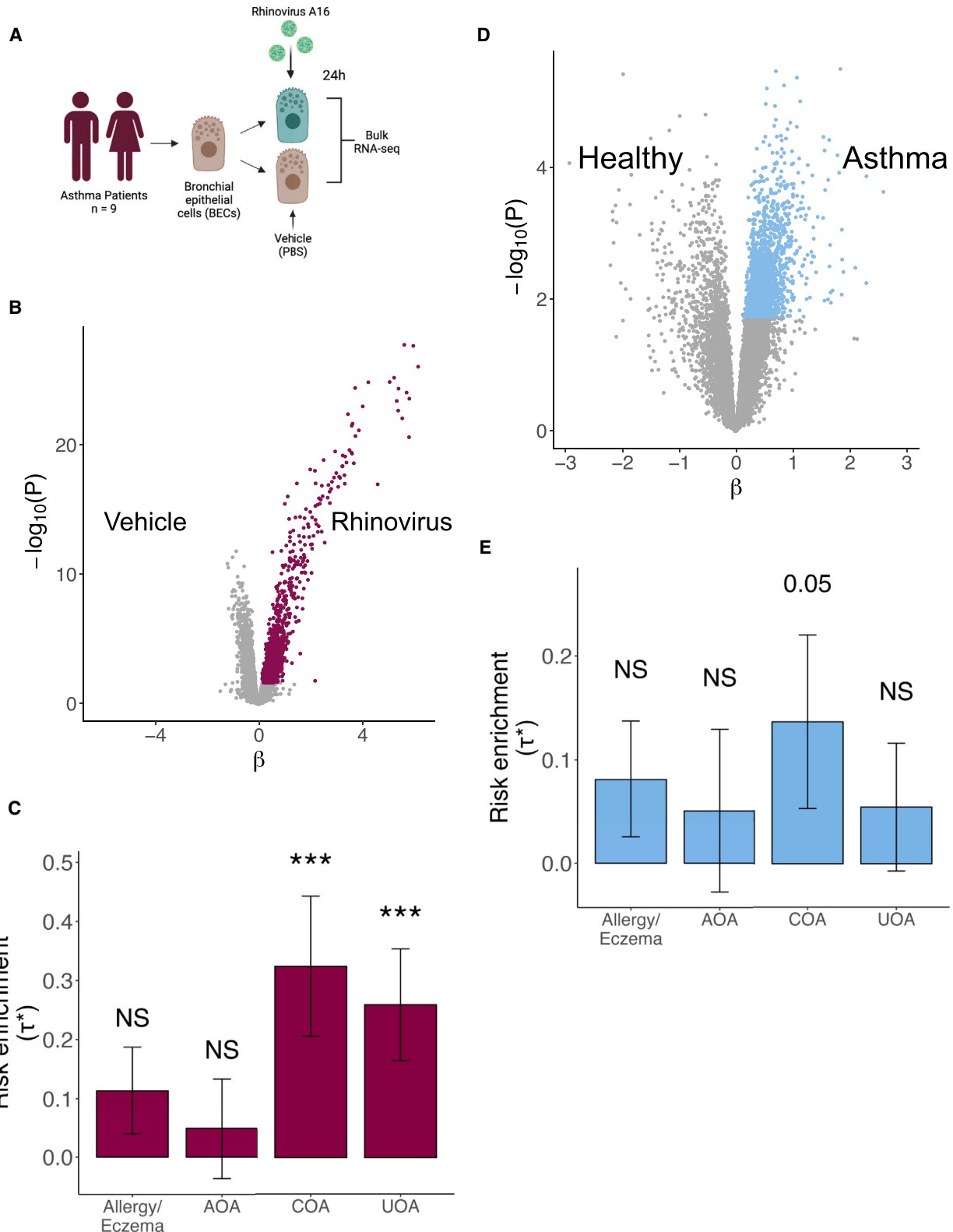

(legend on next page)

## Enrichment of asthma-associated genes after RV infection is stronger in epithelial cells from asthma patients

Having observed the enrichment of asthma-associated genes after RV infection in both bulk and single-cell level in healthy subjects, we asked whether we would observe the same enrichment in samples coming from asthma patients. To do this, we repeated the DEA between RV-A16 infection and treatment with PBS from the first dataset,[41] this time using the asthma patient cohort (Figure 3A). We found 2,843 differentially expressed genes at 5% FDR, 1,353 of them upregulated and 1,481 downregulated after RV infection (Figure 3B and Table S3). Of the 2,843 differentially expressed genes in patients, 1,834 were also differentially expressed in healthy controls. After selecting the top 10% genes by t-statistic (1,488) and running LDSC-SEG, we observed significant heritability enrichment for UOA ($p = 0.003$) and COA ($p = 0.003$; Figure 3C and Table S2). We did not observe heritability enrichment for downregulated genes by RV ($p > 0.05$; Figure S7A and Table S2). While the results were consistent with what we observed in healthy controls (COA $\tau^* = 0.17$, UOA $\tau^* = 0.15$), the enrichment of RV-upregulated asthma-associated genes was more significant and had a larger enrichment coefficient in the asthma patients (COA $\tau^* = 0.32$, UOA $\tau^* = 0.25$).

Based on these results, we hypothesized that asthma patients might have airway epithelial cells in a transcriptomic state that over-expresses asthma-risk genes in comparison to healthy controls, which might be linked to or independent of their response to RV. To test this, we analyzed differentially expressed genes between asthma patients and healthy individuals taking all samples while controlling for RV/PBS treatment. We found 994 differentially expressed genes between patients and controls (5% FDR; Figures 3D and Table S3). After selecting the top 10% genes upregulated in patients based on t-statistic (1,593), we found a suggestive significant enrichment for COA heritability ($p = 0.05$; Figure 3E and Table 2). As expected, our control traits did not have any significant heritability enrichment for either of the annotations tested (Figures S7B–S7C and Table 2). Overall, these results suggest that the epithelial cells from patients could be in a state that is over-expressing asthma-associated genes.

## Likely target genes at asthma-risk loci are upregulated in airway epithelial cells after RV infection

We then investigated which of the genes upregulated by RV infection are associated with COA and AOA. To do so, we retrieved the GWAS lead variants identified by Ferreira et al.[15] We then linked risk variants to genes using three approaches: (1) selecting the likely target genes identified by the locus-to-

gene (L2G) algorithm of Open Targets Genetics, herein called L2G genes; (2) selecting the closest gene to the lead variant; and (3) selecting genes within a 250-kb window of the lead variant (see STAR Methods).[50] From these gene lists, we selected genes that were upregulated upon RV infection in epithelial cells at 5% FDR (Figure 4A).[41,42] For COA, we identified 55 risk loci with genes upregulated upon RV infection in epithelial cells, 13 of which have L2G likely target genes (e.g., *IL1RL1*, *IL4R*, *GSDMB*, *OVOL1*, *MYC*) and six are the closest gene to the lead variant (e.g., *IRF1*, *GPR183*). For AOA, 19 risk loci have genes upregulated by RV in epithelial cells, among which three are likely target genes (*IL4R*, *HDAC7*, *IL1RL1*) and three are the closest gene to the lead variant (*RAPGEF3*, *IRF1*, *SSR3*). Some genes were shared between COA and AOA (*IRF1*, *IL4R*, *PDLIM4*, *IL1R2*, and *IL1RL1*).

After having characterized the asthma-associated genes that are upregulated in RV-infected epithelial cells, we sought to define which of these genes are shared with T cell activation. To do so, we compared the levels of expression of the genes in the RV datasets with a dataset we previously published consisting of eight activation time points of human periphery memory CD4$^+$ T cells stimulated with anti-CD3/CD28 microbeads.[31] In this T cell dataset, we identified genes that were upregulated at 24 h post-stimulation when compared to resting at 5% FDR. We found that one of the highlighted genes that also had an increased expression in T cells was *MYC* ($p = 7.41e-42$), with an observable increase starting at 2 h post stimulation. Notably, this gene's expression was not only increased after RV infection within asthma patients ($p = 0.01$) and within healthy controls ($p = 0.0009$) but was also upregulated in patients compared to controls ($p = 0.01$). On the other hand, *OVOL1*, another GWAS gene upregulated in RV-infected epithelial cells, shows the same pattern of expression as *MYC* in epithelial cells but shows almost no expression in T cells (Figures 4B and S8). *OVOL1* is also associated with atopic dermatitis, another type 2 inflammatory disease, and a recent meta-analysis study confirmed this susceptibility locus for eczema-associated asthma.[51] Other RV-induced asthma-associated genes in epithelial cells that are also upregulated in activated T cells are reported in Tables S6 and S7, and those that are not significantly upregulated in activated T cells are reported in Tables S8 and S9.

## Not all viral infections significantly upregulate asthma-associated genes in epithelial cells

We next asked whether other viruses could potentially be inducing upregulation of asthma-associated genes in epithelial cells. First, we analyzed a bulk RNA-seq dataset of BECs stimulated by

---

**Figure 3. RV-infected BECs from asthma patients show a strong enrichment for asthma risk**

(A) Experimental design of Helling et al.[41] dataset consisting of BECs from asthma patients stimulated with PBS or RV-A16.

(B) Volcano plot showing differentially expressed genes after RV infection in BECs from patient samples. Genes upregulated after infection were selected based on t-statistic and are colored in burgundy.

(C) Bar plot showing LDSC-SEG heritability enrichment coefficient ($\tau^*$) across asthma-related diseases. Error bars represent $\tau^* \pm$ standard error. ***Bonferroni-adjusted $p < 0.05$; NS, nonsignificant ($p > 0.05$).

(D) Volcano plot showing differentially expressed genes between asthma patients and healthy donors. Genes upregulated in patients were selected based on t-statistic and are colored in blue.

(E) Bar plot of LDSC-SEG heritability enrichment coefficient ($\tau^*$). Error bars represent $\tau^* \pm$ standard error. NS, nonsignificant ($p > 0.05$).

See also Figure S7 and Tables S2 and S3.

## B

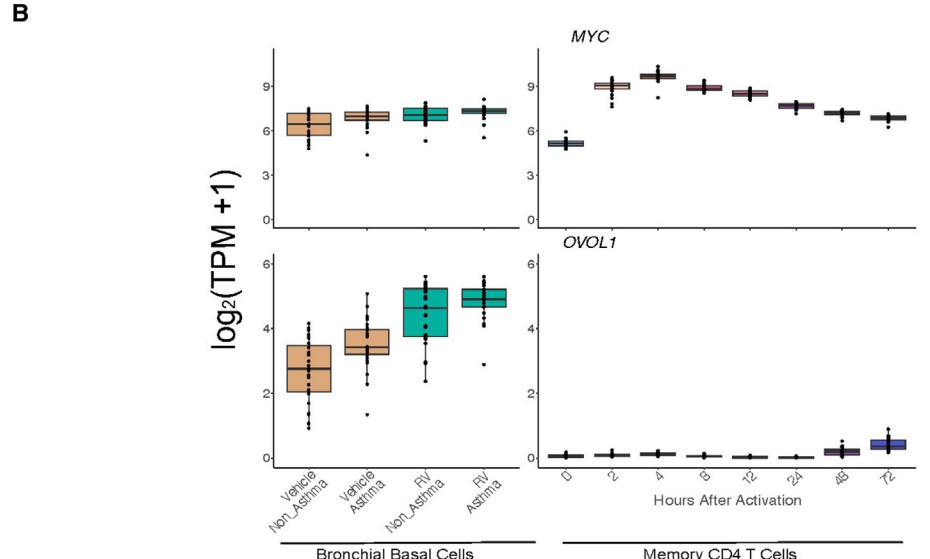

influenza A virus at 48 h or sham control (n = 3 healthy donors; Figure S9A).[52] We identified differentially expressed genes between influenza A and the sham control and we selected the top 10% up-regulated genes ranked by a t-statistic to run LDSC-SEG (Figure S9B and Table S3). The results show a significant enrichment of heritability for the four asthma-related traits (p < 0.05), suggesting that influenza A infection also significantly upregulates asthma and allergy-associated genes (Figure S9C and Table S2). We did not observe any significant enrichment for the control traits (Figure S9D and Table S2).

Subsequently, we analyzed an scRNA-seq dataset of immune and non-immune cells obtained by nasopharyngeal swabs from COVID-19 patients or healthy donors (n = 58; Figures S10A and S10B).[53] We identified cells with significant disease-relevant scores at 10% FDR, among which we found 174 cells for AOA, 795 cells for COA, 594 for allergy/eczema, and 280 for UOA (Figure S10C). COA yielded the highest number of disease-relevant cells. We observed an increase in proportions of enriched cells for T cells in all traits and for squamous cells in COA, allergy/eczema, and UOA (Figure S10D). More precisely, T cells, which represented 5% of the cells in the dataset, constituted 78% of significant cells for AOA, 11% for COA, 42% for allergy/eczema, and 61% for UOA (Figure S10D). The COVID-19 status did not have any significant impact on the asthma-associated expression, as the disease-relevant cells were not over-represented in any specific patient group (Figure S10E). As expected, we observed an enrichment for the macrophages cluster in Alzheimer's and T cells cluster in RA (Figure S10F). Additionally, we analyzed a dataset of BECs infected with severe acute respiratory syndrome coronavirus 2 (SARS-CoV-2) in vitro, and profiled with scRNA-seq after 1, 2, or 3 days, along with non-infected cells (n = 1 healthy control; Figures S11A and S11B).[54] We clustered and annotated ciliated and non-ciliated cell subsets and applied scDRS for the four asthma-related traits (Figures S11C and S11D). We identified cells with significant disease-relevant score at 10% FDR, finding three cells for AOA, 802 cells for COA, 243 cells for allergy/eczema, and 621 cells for UOA (Figures S11D and S11E). The disease-relevant cells were strongly over-represented in the non-infected cells subsets, constituting 62%–89% of the significant cells (Figure S11F). Overall, these findings indicate that the influenza virus significantly induces the expression of asthma-associated genes, whereas SARS-CoV-2 does not.

### Stimulation with type 2-related cytokines does not induce significant upregulation of asthma-associated genes in epithelial cells

Both epithelial and immune cells respond to cytokines by upregulating signaling pathways that drive inflammation. Some pro-inflammatory cytokines relevant to asthma are interleukin (IL)-4, IL-13, IL-17, and IFNγ. These cytokines are upregulated in subsets of patients with severe or type 2 asthma.[55–60] Moreover, blockade of IL4Rα is a highly effective treatment for moderate to severe asthma.[61] Consequently, we asked whether airway epithelial cells stimulated with cytokines might induce a transcriptional program enriched for asthma-associated genes. First, we used a bulk RNA-seq dataset consisting of human BECs that were stimulated with either IFNα, IFNγ, IL-13, or IL-17 (n = 6 healthy donors; Figure 12A).[62] We selected the top 10% upregulated genes by t-statistic for each stimulus (Figure 12B and Table S3). We used LDSC-SEG to analyze these four sets of genes in the four asthma-associated traits and the control traits. We did not identify any significant heritability enrichment for any of the stimuli gene sets for the asthma-associated traits tested here (Figure S12C and Table S2), nor for the control traits (except for genes upregulated by IFNγ for RA; Figure S12D and Table S2).

Next, given that IL-13 might work synergistically with IL-4, we performed bulk RNA-seq of nasal airway epithelial cells from healthy donors (n = 5) co-stimulated in vitro with IL-4 and IL-13 (Figure S13A). We tested DE genes for the IL-4-IL-13 condition compared to the unstimulated control (Figure S13B and Table S3). In line with the results observed in the previous analysis, we found no significant enrichment for any of the asthma-associated (Figure S13C and Table S2) or control traits (except for AD; Figure S13D and Table S2). Together, these results suggest that epithelial cells upregulate asthma-associated genes in a stimulus-specific manner, which, to the extent of this study, is not caused by the stimulation with the cytokines tested here.

## DISCUSSION

While some genetic risk variants for asthma are enriched near genes with T cell-specific expression,[20,23,28,34] the effects of most variants on gene regulation remain unknown. In this study, we asked whether some of these missing regulatory effects could be hidden in airway epithelial cells, given they are the first line of contact for respiratory viruses, including those that have been associated with asthma development or exacerbations. We analyzed 10 transcriptomic datasets of human airway epithelial cells cultured under different stimuli and integrated them with genetic susceptibility data for asthma and related traits. We consistently showed that RV-activated epithelial cells significantly upregulate genes at COA risk loci. We observed this in samples from healthy donors and even more so in cells from asthma patients. Notably, we discovered that non-ciliated cells are the subset driving these associations with asthma, indicating

**Figure 4. Genes in COA and AOA risk loci that are upregulated with RV in airway epithelial cells**
(A) Miami plot of COA and AOA GWAS. Each gray dot shows the SNP found in the Ferreira et al.[15] GWAS. The black circles represent SNPs being found as lead variants in either Open Targets Genetics or in the Ferreira et al.[15] study. Genes shown are those that get significantly upregulated in airway epithelial cells after RV infection: in purple are the likely target genes identified by the L2G Open Targets algorithm, in blue are the closest genes to the lead variant, and in black are genes found within 250 kb of the lead variant. The blue dashed line represents the p-value threshold of −log10(5 − 10⁻⁸).
(B) Boxplots depicting gene expression levels for *MYC* (top) and *OVOL1* (bottom). Left panel shows gene expression in BECs from the Helling et al.[41] dataset; samples infected with RV-A16 are shown in green and non-infected samples are in brown. Right panel shows gene expression from the Gutierrez-Arcelus et al.[31] dataset of human CD4 memory T cell activation time course; colors depict time points after activation. In each plot, every point represents an individual sample. See also Figure S8 and Tables S6–S9.

that non-ciliated airway epithelial cells activated with RV are key mediators of genetic susceptibility to COA. While other respiratory viruses, such as influenza might also significantly upregulate genes at asthma-risk loci, this is not likely a general virus response or epithelial cell activation signature, given that we did not detect asthma heritability enrichment for SARS-CoV-2 or cytokine-upregulated genes.

Our findings are consistent with epidemiological studies that have shown associations between wheezing illness caused by RV infection and asthma development in children.[4,63–65] Additionally, a previous birth cohort study identified genetic variants at the 17q21-q12 locus that were associated with asthma in children who had RV-associated wheezing illness in the first 3 years of life, but not in children who had respiratory syncytial virus-associated wheezing illnesses at those same ages.[9] In that study, RV upregulated two genes at this locus, *ORMDL3* and *GSDMB*, in PBMCs.[9] Here, we observe that, in non-ciliated airway epithelial cells, RV induces upregulation of *GSDMB* as well as putative causal genes in 54 additional risk loci. This demonstrates a widespread interaction between *in vitro* RV infection and polygenic susceptibility to COA, specifically mediated through airway epithelial cells. In contrast, 19 risk loci for AOA have genes that get upregulated with RV infection in airway epithelial cells. This result is in part due to the lower number of genome-wide significant loci in AOA (56) compared to COA (126), even if the AOA GWAS had a larger sample size. In fact, AOA presented positive heritability enrichment coefficients for RV-upregulated genes in airway epithelial cells (Figures 1 and 3); however, for most analyses, the heritability enrichment was not statistically significant. This contrasts with the T cell-specific annotations, for which AOA had statistically significant enrichments in most analyses. Hence, our data suggest that RV-infected airway epithelial cells may play a more important role in mediating genetic susceptibility to COA compared to AOA. These findings are concordant with a previous study reporting that genes at COA-specific risk loci (as compared to AOA) have high expression in skin, which is a barrier tissue with an abundance of epithelial cells.[14] Overall, our findings support the hypothesis that RV could be causally linked to asthma development in children and not just be a biomarker of children destined to develop asthma. Not all children that get RV wheezing develop asthma, and our findings suggest that the combination of preschool RV wheezing illnesses and a high genetic burden synergistically promote the development of childhood asthma.

Our study highlighted genes in GWAS loci that get upregulated with RV in airway epithelial cells. Twelve of these are within the list of leading prioritized genes as potential drug targets reported by a recent study integrating asthma GWAS information with protein-protein interaction data (*IL-6, MYC, PRKCQ, ETS1, IL-4R, IRF1, IL-1R2, RELA, CDK2, SOCS1, NFKB2, PSMA6*).[66] Of these, IL-6 is a target for a therapy that is in phase II clinical trials to treat severe asthma (clazakizumab), and IL-4R is a target for dupilumab, which has been tested in clinical trials for uncontrolled severe asthmatics.[67] Our study underscores the possibility that these therapies might in part act through airway epithelial cells and raises the consideration of future therapies being designed as to specifically deliver to airway epithelial cells and/or

when genetically susceptible subjects are infected with RV. In fact, GSK has a phase I clinical trial on an inhalation powder for treatment during RV infection in subjects with mild asthma.[68]

We discovered that non-ciliated cells (basal, secretory, and transitional) are the specific cell subsets that overexpress genes at asthma-risk loci. This suggests that an important fraction of the non-coding risk variants for asthma likely affect gene regulation in non-ciliated cells under specific viral activation states. In our study, we looked at two different RV types. For the case of RV-C15, the receptor of the virus, CDHR3, is mainly expressed in the ciliated cells,[69,70] and viral RNA quantification confirmed this subset is the one directly infected by the virus (Figure S5F). This suggested that RV-infected ciliated cells efficiently transmit a signal to the non-ciliated cells, which then express genes in asthma-risk loci. The time-course experiments indicated this upregulation of asthma-associated genes occurred predominantly at 24 and 42 h.[42] In our analyses of RV-A16-infected epithelial cells, the data came from bulk RNA-seq of basal cells (non-ciliated) treated for 24 h. In contrast to RV-C15, RV-A16 binds to the intercellular adhesion molecule (ICAM) receptor, which is expressed in ciliated cells and basal cells.[71] Strikingly, while the cell subsets that get directly infected differ between the two RV strains, both significantly upregulated asthma-associated genes in non-ciliated cells at 24 h. By contrast, for SARS-CoV-2, *in vitro* infection did not upregulate genes in asthma-risk loci; rather, the non-infected cells presented a significant expression of asthma-associated genes. Although the data in this study came from only one individual, the cells with significant disease-relevant scores were also predominantly non-ciliated cells (>83%; Figure S11). Future single-cell studies with larger sample sizes and ascertaining infection by multiple types of viruses could point to additional epithelial cell subsets and cell states as candidate drivers of genetic susceptibility to asthma.

Another cell type that showed significant over-expression of genes in COA risk loci was the myeloid lineage in nasal samples from chronic rhinosinusitis patients and healthy controls[39] (Figure S2). This is in line with the study of macrophages in the context of asthma[72,73] and with the findings of a recent large-scale scRNA-seq study in lung where they reported co-localization between COA risk loci and eQTLs in macrophages, monocytes, and dendritic cells. Together, these findings suggest myeloid cells from the respiratory tract could be another important cell type worth studying further to find new genetic mechanisms for COA.

The observations in our study may also be relevant to virus-induced asthma exacerbation.[74] Here, we not only demonstrate that RV infection induces a transcriptional response enriched in COA risk but we also identified a heritability enrichment for genes upregulated in asthma patients compared to controls, even when controlling for RV infection (suggestive $p = 0.051$; Figure 3E). This result is in line with a previous observation that, at the open-chromatin level, airway epithelial cells of asthma patients have a large amount of open-chromatin regions at baseline that are RV-response regions in healthy controls.[41] It is possible that over-expression of asthma-associated genes at baseline may increase the risk for acute virus-induced exacerbations in patients with asthma. Influenza infections, which can cause asthma exacerbations (especially in adults),[75–77] induced an

enrichment of both adult-onset and COA heritability in influenza-upregulated genes in airway epithelial cells. Furthermore, SARS-CoV-2 seems less likely than other viruses to provoke asthma exacerbations, and asthma does not appear to be a risk factor for severe SARS-CoV-2 infection.[78] This could be due to multiple reasons, such as allergy-induced reduction in the ACE2 receptor,[79] but it is also in line with our observations that SARS-CoV-2 infection itself does not induce a transcriptional program significantly enriched in asthma heritability.[79]

## Limitations of the study

Our study had some limitations. We were not able to ascertain all possible epithelial cell subsets and states. Most of the datasets analyzed involved *in vitro* infections, rather than *in vivo*-infected samples. Additionally, all samples came from adults, which made it all the more striking that we detected heritability enrichments for COA. Future studies in children are important to validate these findings. Furthermore, we were limited by the cell sources and specific time points and experimental designs of each study. In particular, the absence of asthma heritability enrichment in cytokine-upregulated genes in epithelial cells could imply multiple scenarios. One possibility could be that, even though pro-inflammatory cytokines (IL-4, IL-13, IFNα, IFNγ, IL-17) upregulate many genes in epithelial cells (684–2,876 at 5% FDR in our analyses), they do not significantly interact with polygenic risk factors for asthma in epithelial cells. Other possibilities for the absence of signal could be that the cytokine-induce activation might interact with genetic risk factors acting in T cells or other non-epithelial cells, or that there are interactions with environmental conditions not present in the models included in this analysis. Another limitation of our study is that it primarily focuses on GWAS data derived from European-ancestry individuals, partly due to the bias in GWAS studies so far,[80] and consequently there are limitations posed by the tools utilized in our study, which predominantly leverage data from specific populations. A recent multi-ancestry meta-analysis for asthma highlighted that genetic effects are largely consistent between ancestries but that ascertaining all ancestries is important to find all of the risk loci for asthma, as there are risk alleles that are frequent in some populations but not others.[11] Broadening the scope of heritability enrichment analyses to incorporate multi-ancestry meta-analyses would help to further characterize the cell types and cell states relevant for asthma and its related traits. Moreover, given that the cases in the GWAS utilized in this study are based on self-reported doctor-diagnosed disease and PheCodes from the UK Biobank, inaccuracies in patient classification are expected, and replication of findings using summary statistics from future GWASs involving better-characterized patients and controls is important.

## Conclusions

Overall, our findings of asthma heritability enrichment in various epithelial cell states (resting versus virus infected, patients versus healthy controls) could reflect variability in how risk variants contribute to disease onset versus progression. These results highlight the importance of studying the cellular context in which GWAS loci contribute to disease risk and will ultimately help to better understand the mechanisms through which those risk variants are acting. Moreover, the outcomes of our study could open the door to new therapeutic avenues. Indeed, drug targets that have genetic evidence are more likely to be approved and move forward to clinical trials than those without it.[81] Large-scale multi-omic studies (with comparable power to GWASs[82]) of non-ciliated airway epithelial cells activated with RV could help identify the target genes of non-coding asthma-risk variants, together with functional validations with approaches such as base editing. Finally, if our observations are confirmed and further characterized by future studies, it would support the development of an RV vaccine or other protective intervention as a way to prevent COA.[83]

## STAR★METHODS

Detailed methods are provided in the online version of this paper and include the following:

- KEY RESOURCES TABLE
- RESOURCE AVAILABILITY
  - Lead contact
  - Materials availability
  - Data and code availability
- METHOD DETAILS
  - GWAS collection
  - Air-liquid-interface (ALI) culture of human BECs
  - Summary statistics processing for visualization
  - Bulk RNA-seq data processing and quality check
  - Differential expression analyses
  - ATAC-seq data processing and differential accessibility analysis
  - Single-cell RNA-seq, QC, and analysis
  - LDSC-SEG
  - MAGMA and scDRS
  - Cell-cell interaction between ciliated epithelial cells and non-ciliated epithelial cells
  - Linking variants to genes
  - Software description (plots, R, and Biorender)

## SUPPLEMENTAL INFORMATION

## ACKNOWLEDGMENTS

This study was supported by the National Institutes of Health (NIH) grant U19AI095219. C.O. was supported by NIH grant U19AI162310. N.A.B. was supported by NIH grants AI134989 and U19AI095219. J.E.G. and N.A.B. were supported by NIH AADCRC RNA Sequencing Core for Airway Epithelial Cells. M.G.-A. was supported by NIH grants P30AR070253 and U19AI095219. We thank Donata Vercelli, Luis Barrera, Peter Nigrovic and his laboratory, and the Gutierrez-Arcelus laboratory for feedback on this study. This work was supported by the Cell Discovery Network, a collaborative initiative funded by The Manton Foundation and The Warren Alpert Foundation at Boston Children's Hospital.

## AUTHOR CONTRIBUTIONS

Conceptualization, S.D., D.F.-S., N.A.B., J.E.G., and M.G-A.; methodology, S.D, D.F.-S, S.G., and M.G.-A.; software, S.D. and D.F.-S.; formal analysis, S.D., D.F.-S., and C.M.; resources, C.O., J.E.G., and N.A.B.; writing – original draft S.D., D.F.-S., and M.G.-A.; writing – review & editing, S.D., D.F.-S., V.R.C.A., G.X.H., C.M., C.O., S.G., J.A.B., N.A.B., J.E.G., and M.G.A.; visualization, S.D., D.F.-S., and M.G.-A.; supervision, G.X.H., V.R.C.A., C.K., S.G.,

J.A.B., J.E.G., N.A.B., and M.G.-A.; funding acquisition, J.A.B., N.B., J.E.G., and M.G.-A.

**DECLARATION OF INTERESTS**

J.A.B. has served on scientific advisory boards for Siolta Therapeutics, Third Harmonic Bio, and Sanofi/Aventis. N.A.B. has served on scientific advisory boards for Regeneron.

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

## STAR★METHODS

### KEY RESOURCES TABLE

| REAGENT or RESOURCE | SOURCE | IDENTIFIER |
|---|---|---|
| **Deposited data** | | |
| ATAC-seq of 25 blood cell types from healthy donors | Calderon et al.[22] | GEO: GSE118189 |
| scRNA-seq of nasal samples from chronic rhinosinusitis patients | Wang et al.[39] | Genome Sequence Archive:HRA000772 |
| scRNA-seq of allergen-exposed T cells from patients with allergy and asthma | Seumois et al.[40] | GEO: GSE146170 |
| RNA-seq of BECs infected with RV-A for asthma cases and non-asthma controls | Helling et al.[41] | GEO: GSE152550 |
| RNA-seq of airway epithelial cells infected with RV-C | Basnet et al.[42] | ImmPORT: SDY1882 |
| scRNA-seq of airway epithelial cells infected with RV-C | Basnet et al.[42] | ImmPORT: SDY1882 |
| RNA-seq of BEC infected with influenza A | Tao et al.[52] | GEO: GSE193164 |
| scRNA-seq of SARS-CoV-2 infection in human airway epithelium | Ravindra et al.[54] | GEO: GSE166766 |
| RNA-seq of BECs with and without stimulation of IFN-a, IL-17, and IL-13 | Koh et al.[62] | GEO: GSE185200 |
| RNA-seq of airway epithelial cells stimulated with IL-4 IL-13 | This study | GEO: GSE268072 |
| RNA-seq of CD4[+] memory T cells activated with anti-CD3/CD28 | Gutierrez-Arcelus et al.[31] | GEO: GSE140244 |
| Adult-onset asthma summary statistics | Ferreira et al.[15] | https://www.ebi.ac.uk/gwas/studies/GCST007799 |
| Childhood-onset asthma summary statistics | Ferreira et al.[15] | https://www.ebi.ac.uk/gwas/studies/GCST007800 |
| Allergy/eczema summary statistics | UK Biobank Sudlow et al.[84] | https://console.cloud.google.com/storage/browser/broad-alkesgroup-public-requester-pays/UKBB |
| Unspecified-onset asthma summary statistics | UK Biobank Sudlow et al.[84] | https://console.cloud.google.com/storage/browser/broad-alkesgroup-public-requester-pays/UKBB |
| Height summary statistics | Lango et al.[85] | https://console.cloud.google.com/storage/browser/broad-alkesgroup-public-requester-pays/UKBB |
| Alzheimer's disease summary statistics | Lambert et al.[86] | https://console.cloud.google.com/storage/browser/broad-alkesgroup-public-requester-pays/UKBB |
| Rheumatoid arthritis summary statistics | Okada et al.[87] | https://console.cloud.google.com/storage/browser/broad-alkesgroup-public-requester-pays/UKBB |
| scRNA-seq study of nasopharyngeal swabs from donors with and without COVID-19 | Ziegler et al.[53] | https://singlecell.broadinstitute.org/single_cell/study/SCP1289/impaired-local-intrinsic-immunity-to-sars-cov-2-infection-in-severe-covid-19 |
| **Software and algorithms** | | |
| STAR (v2.7.9a) | Dobin et al.[88] | https://github.com/alexdobin/STAR |
| Salmon tool (v1.5.1) | Patro et al.[89] | https://github.com/COMBINE-lab/salmon |
| RSEM (v1.3.3) | Li and Dewey[90] | https://github.com/deweylab/RSEM |
| cellranger count and aggr(v6.1.2) | 10X Genomics | https://www.10xgenomics.com/support/software/cell-ranger/latest/analysis/running-pipelines/cr-gex-count; https://www.10xgenomics.com/support/software/cell-ranger/latest/analysis/running-pipelines/cr-3p-aggr |
| Seurat (v4.0.5) | https://doi.org/10.1016/j.cell.2021.04.048 | https://satijalab.org/seurat/ |
| Harmony (v0.1.1) | Korsunsky et al.[91] | https://github.com/immunogenomics/harmony |
| MCPcounter (v1.2.0) | Becht et al.[92] | https://github.com/ebecht/MCPcounter |

*(Continued on next page)*

*Continued*

| REAGENT or RESOURCE | SOURCE | IDENTIFIER |
|---|---|---|
| Bedtools (v2.31) | Quinlan et al.[93] | https://github.com/arq5x/bedtools2 |
| LDSC-SEG | Finucane et al.[20] | https://github.com/bulik/ldsc/wiki/Cell-type-specific-analyses |
| scDRS (v1.0.2) | Zhang et al.[23] | https://github.com/martinjzhang/scDRS |
| MAGMA (v1.10) | de Leeuw et al.[94] | https://github.com/martinjzhang/scDRS/issues/2 |
| CellphoneDB (v5) | Efremova et al.[46] | https://github.com/Teichlab/cellphonedb |
| GitHub (original codes supporting this work) | https://zenodo.org/doi/10.5281/zenodo.12775392 | https://github.com/gutierrez-arcelus-lab/Asthma_EpithelialCells/tree/v1 |

## RESOURCE AVAILABILITY

### Lead contact
Further information and requests for resources and reagents should be directed to and will be fulfilled by the lead contact, Maria Gutierrez-Arcelus (mgutierr@broadinstitute.org).

### Materials availability
This study did not generate new unique reagents.

### Data and code availability
Raw files and counts matrix of RNA-seq generated in this study have been deposited at Gene Expression Omnibus (GEO) and are publicly available. Accession numbers are listed in the key resources table. All other datasets are publicly available data. These accession numbers for the datasets are listed in the key resources table. The scripts to perform differential expression and to run LDSC-SEG and scDRS have been deposited in a public repository available at https://github.com/gutierrez-arcelus-lab/Asthma_EpithelialCells and at Zenodo. DOIs are listed in the key resources table. Any additional information required to reanalyze the data reported in this paper is available from the lead contact upon request.

## METHOD DETAILS

We downloaded transcriptomic datasets from the National Center for Biotechnology Information (NCBI), GEO, Genome Sequence Archive (GSA) and from ImmPORT. We also downloaded a chromatin accessibility dataset (ATAC-seq) from GEO.

### GWAS collection
We downloaded pre-processed summary statistics for the four asthma-associated traits; adult-onset asthma, childhood-onset asthma, unspecified-onset asthma, allergy/eczema and for those in our control panel; height, Alzheimer's disease and rheumatoid arthritis (Table S1). We downloaded the pre-processed summary statistics from Alkes Price laboratory website: https://console.cloud.google.com/storage/browser/broad-alkesgroup-public-requester-pays/UKBB. The adult-onset and childhood-onset asthma are GWAS with careful curation of the UK Biobank data and therefore include fewer individuals (case/control numbers in Table S1, for details on patient selection criteria please see Ferreira et al; [15]). The adult-onset asthma GWAS was performed with patients with age at first diagnosis between 20 and 60 years of age. The childhood-onset asthma GWAS was performed with patients with age at first diagnosis of 19 years of age or younger. The unspecified onset-asthma GWAS was performed with less meticulous curation of UK Biobank data and includes individuals with any age of onset.[84]

### Air-liquid-interface (ALI) culture of human BECs
Air-liquid interface cultures were grown from nasal basal epithelial cells from 6 healthy adult donors. ALIs were allowed to mature for 14 days, then stimulated with 10 ng/mL of IL-4 and 10 ng/mL of IL-13 for an additional 7 days, and then lysed with TCL buffer (Qiagen 1031576) at the conclusion of the experiment. Lysates were stored at −80C and later submitted to the Broad for SmartSeq2 low input bulk RNA-seq (38bp paired-end sequencing).

### Summary statistics processing for visualization
For visualization of GWAS SNPs in Figure 4, we downloaded the childhood-onset asthma and adult-onset asthma summary statistics from the GWAS catalog (GCST007800, GCST007799). We used the harmonized summary statistics in the GRCh38 version of the genome. We removed the MHC region (chr6:28510120-33480577).

### Bulk RNA-seq data processing and quality check

FASTQ files were aligned to the GRCh38 or GRCh37 human genome using STAR (v2.7.9a) with standard parameters and two-pass mode, or the Salmon tool (v1.5.1). For BAM files generated with STAR, counts were calculated using RSEM (v1.3.3). We normalized the counts by transforming them to their log2(TPM+1) value, where TPM stands for transcripts per million. To detect outlier samples, we performed principal component analysis on scaled normalized expression for the top 1000 most variable genes that were expressed in at least 25% of the samples. For alignment and quantification, we used the ENSEMBL reference annotation release 105 which was downloaded from the ENSEMBL website.

### Differential expression analyses

Differential gene expression was tested using a linear mixed model, similar to what we did in Gutierrez-Arcelus et al.[95] Specifically, we used a likelihood ratio test between two nested models (*anova* function in R). In these models, gene expression levels (log2(TPM +1)) represent the dependent variable. "Donor ID" was included as a predictor variable, treated as a random effect. To compare one condition against the others, we indicated with 1 the tested condition and 0 for the others (the test variable). We used the function "lmer" from the R package "lme4" to implement the model. For risk gene visualization in the miami plot, P-values were corrected for multiple hypothesis testing using the package "qvalue". Differentially expressed genes at 5% FDR are reported for depicting specific genes in risk loci. After each analysis, we calculated a t-statistic for each gene to rank them and chose the top 10% as annotations for heritability enrichment analysis (see LDSC-SEG section below). The details of each analysis are divided by dataset and described in the following section.

#### Helling dataset

To assess differentially expressed (DE) genes between rhinovirus treatment and phosphate buffered saline (PBS) vehicle control within healthy individuals, we tested genes that had a normalized count greater than 1 in at least 9 samples which led to a total of 14,883 genes. The threshold for the minimum number of samples reflected half of the biological replicates (9/18). In addition to having "donor ID" as a random effect, we accounted for "sex" as a fixed effect. We repeated this process in asthma patients only, testing 14,888 genes for differential expression between rhinovirus infection and PBS vehicle control. To find DE genes from asthma patients compared to healthy controls, we took all the samples and recalculated the number of genes present in at least 9 samples. We tested 15,935 genes and incorporated "treatment" with either PBS or rhinovirus as a fixed effect covariate in the model, and tested for disease status.

#### Tao dataset

We included 16,031 genes having a normalized count greater than 1 in half of the samples (3/6 samples). We tested for differential expression between influenza treatment and control (sham).

#### Koh dataset

We tested 14,988 genes, to find differentially expressed genes specific to each condition compared to all others: IFNα, IFNγ, IL-13, and IL-17, respectively. We selected genes having a normalized count greater than 1 in at least 6 samples. This number reflected the smallest amount of replicates found across conditions (6/36 samples).

#### Basnet dataset

We excluded the resting sample from donor B03 from these analyses (see Methods, Bulk RNA-seq, QC, and analysis). To obtain differentially expressed gene profiles for each time point, we performed four separate models in which we tested a single time point against the other two and the resting condition. Our fourth model tested for DE genes in activation conditions versus the resting state through the same approach. Since we had 3 biological replicates for most time points, we tested genes with a normalized count greater than 1 in at least (3/11) samples, which yielded 14,380 genes. This gene set was used for all models.

#### Gutierrez-Arcelus dataset

We perform differential expression between stimulated T cells for 24 h (with anti-CD3/CD28 beads) against resting condition with the 13,061 genes that had a normalized count greater than 1 in at least 36 of the samples (N donors = 24). We included donor as a random effect.

#### Human BECs co-stimulated with IL-4 and IL-13

To obtain genes DE under IL-4 and IL-13 co-stimulation we compared them against the non-stimulated cells. We tested 13,518 genes that had a normalized count greater than 1 in at least half of the samples (9/18) samples.

### ATAC-seq data processing and differential accessibility analysis

We used the 829,942 consensus peaks called by Calderon et al.[22] (peaks were called in each sample separately, then merged across samples, and then counts were re-calculated for all samples using the merged peak coordinates). We transformed counts into reads per kilobase per million (RPKM), then quantile normalized and finally scaled to their log2(normalized RPKM+1). To assess differentially accessible (DA) peaks, we first calculated the mean normalized count per cell type and then created cell-type accessible sets of peaks. We included a peak in the set if it had a normalized count greater than the mean of the cell type in at least half of the samples corresponding to subtypes of that cell type. We tested between 400 and 600 thousand peaks per cell-type for DA. To do so, we implemented a linear mixed model using the normalized counts as the response variable, and for the predictor, a bit flag system

## Article

indicating 1 if the sample belonged to the tested cell type and 0 for the remaining cells. Peaks were sorted by t-statistic and we took the top 10% peaks for each cell-type-specific annotation. This process was replicated for a second selection model, implemented to divide peaks between stimulated and unstimulated categories.

### Single-cell RNA-seq, QC, and analysis

FASTQ files from the single-cell RNA-seq dataset from Wang et al., Seumois et al., Basnet et al., and Ravindra et al.,[39,40,42,54] were downloaded from the database indicated in the key resources table. For each dataset, we aligned FASTQ files to the human reference human genome GRCh38,[96] using the GENCODE release 32 with cellranger count (v6.1.2), using default parameters. For Basnet et al.,[42] we added the RV-C15 sequence to the reference genome. Counts were then aggregated using the cellranger aggr (v6.1.2) function with the default parameters. The subsequent analyses were done for each of the datasets individually. We discarded cells with less than 500 genes expressed and cells expressing more than 20% of mitochondrial genes. We normalized raw counts with the "LogNormalize" method from Seurat package (v4.0.5). We used the normalized counts to perform a PCA with the 1000 most variable genes. We used the top 20 principal components to perform dimensionality reduction with UMAP to visualize the data. We identified clusters using the "FindClusters" function with the Louvain algorithm and a resolution parameter of 0.2, using the top 20 principal components (PCs). We corrected PCs with Harmony package[91] (v0.1.1) as indicated as follows, if nothing is indicated, no corrections were applied. For Wang et al.,[39] dataset we corrected for "donor ID" and "tissue". For Basnet et al., dataset we corrected for "donor ID" and virus infection. For Ravindra et al.,[54] dataset we corrected for virus infection.

To identify which cellular type was present in each cell cluster, we used the function "FindVariableFeatures" (parameter; test.use = wilcox) from Seurat package to identify differentially expressed genes in each cluster. If immune cells were present in the dataset we used the tool MCPcounter (v1.2.0) to annotate immune cells.[92] Based on these cellular markers we annotated the clusters with data from the literature.[42,97,98]

For data from Ziegler et al.[53] we used the UMAP coordinates, and the cell annotation originally published by the authors.

### LDSC-SEG

State-specific gene sets were generated using the top 10% of the genes tested ranked by t-statistic for each of our DE analyses. Genes coordinates were mapped from the human genome reference GRCh37 GTF file and formatted into bed files. We repeated this process to generate control bed files containing all the genes tested for DE in each analysis. Symmetric windows of 100kb were added at each side of the genes using the bedtools (v2.31) "slop" function. For bed files containing ATAC-seq peaks, this window consisted of 225 bp at each side of the peak, to represent a similar genomic coverage. LD-Score files were generated using the LDSC pipeline along with data from HapMap 3 and Phase 3 of the European 1000 Genomes obtained from the LDSC-repository. The regression was run using the baseline model v1.2. We reported the $p$ values of regression coefficients, and normalized regression coefficient as per-standardized-annotation effect sizes $\tau^*$ as in (Gazal et al. 2017 Nat Genet)[99] to allow for multi-trait comparisons. LDSC-SEG uses a one sided t-test to compute regression $p$ values to test if the coefficient is greater than zero. Regression $p$ values were corrected using the "p.adjust" function from R using both FDR and Bonferroni methods. The reference GTF file used to map the genes was obtained from GENCODE (v37).

### MAGMA and scDRS

We used the adult-onset asthma, the childhood-onset asthma, the allergy/eczema, the all-asthma, the rheumatoid arthritis, and the Alzheimer's GWAS summary statistics as well as the corresponding set of 1000 putative disease genes (obtained with MAGMA) provided in the original publication of the scDRS method.[23,94]

We used MAGMA (v1.10) to compute the gene-level association P-values and z-scores from GWAS summary statistics of Height. We transformed P-values to z-scores using this formula: 2*pnorm(abs($Z$ score), mean = 0, sd = 1, lower.tail = F) in R. To map SNPs to genes, we used magma with default parameters specified in the scDRS documentation. We retrieved the top 1000 genes based on MAGMA $Z$ score as putative disease genes.

We used scDRS (v1.0.2) to quantify the expression of the putative disease genes derived from GWAS summary statistics using MAGMA in each cell of each single-cell RNA-seq dataset separately for the 7 GWAS tested and described previously (key resources table). We used the function scDRS "compute-score" with default parameters (–flag-filter-data False). scDRS computes $p$ values per cell based on the empirical distribution of the pooled normalized control scores. We used the R function « qvalue » to correct for multiple hypothesis testing.

### Cell-cell interaction between ciliated epithelial cells and non-ciliated epithelial cells

To identify potential pairs of interactors between RV-infected ciliated and non-infected non-ciliated epithelial cells we used the dataset of Basnet et al.[42] We first retrieved two subsets of the cells contained in this dataset, specifically: RV-infected ciliated cells and non-infected non-ciliated cells. We identified hypothetical pairs of interactors (ligand-receptor) between RV-infected ciliated cells and non-infected non-ciliated cells with CellPhoneDB (v5) with the method "statistical_analysis" (default parameters). We then filtered the results obtained by CellPhoneDB, by retrieving significant ligand-receptor interactions (permutation $p$ < 0.05). We also requested that the ligand be upregulated upon rhinovirus infection within ciliated cells. To evaluate which genes were upregulated between rhinovirus infected ciliated cells and non-infected ciliated cells, we used the Seurat function "FindMarkers" (Wilcoxon test, min.pct = 0.1 and default parameters). We selected upregulated genes based on Log2FC > 0.50 and P adjusted <0.05.

**CellPress**

**Cell Genomics**
Article

## Linking variants to genes

For the Miami plot we used three different and complementary approaches to map variants to genes. First, we used data from Open Target Genetics website.[50,100] We queried the website to retrieve information for childhood-onset asthma (GCST007995) and adult-onset asthma (GCST007799). We then extracted the L2G gene and the Closest Gene for each variant when information was available. We also added a window of 250kb around each variant with the "bedtools slop" function and retrieved the genes falling in those regions with the "bedtools intersect" function. Finally, we obtained the intersect between this "snp-to-gene" list of genes and the genes upregulated upon rhinovirus infection (genes annotated on Figure 4A, see also "differential expression analyses" section).

We also retrieved the lead variants from the original paper from Ferreira et al.[15] We added a window of 250kb around each variant with the "bedtools slop" function and retrieved the genes falling in those regions with the "bedtools intersect" function. We identified the closest gene to the variant by using the function "bedclosest".

## Software description (plots, R, and Biorender)

All the plots were generated with R and graphic schematics were generated with Biorender.

