## [Document S2. Transparent peer review records for Djeddi et al · Cell Genomics]

Rhinovirus infection of airway epithelial cells uncovers the non-ciliated subset as a likely driver of genetic risk to childhood-onset asthma

Sarah Djeddi^{ε,1,2,3}, Daniela Fernandez-Salinas^{ε,1,2,3,4}, George X. Huang^{5,6}, Vitor R. C. Aguiar^{1,2,3}, Chitrasen Mohanty¹⁰, Christina Kendzioriski¹⁰, Steven Gazal^{7,8}, Joshua Boyce^{5,6}, Carole Ober⁹, James Gern^{10,11}, Nora A. Barrett^{5,6}, Maria Gutierrez-Arcelus^{*1,2,3,12}

Summary

Initial submission: Received : 1/27/2024

Scientific editor: Laura Zahn

First round of review: Number of reviewers: 2
Revision invited : 3/17/2024
Revision received : 6/11/2024

Second round of review: Number of reviewers: 2
Accepted : 8/1/2024

Data freely available: Yes

Code freely available: Yes

This transparent peer review record is not systematically proofread, type-set, or edited. Special characters, formatting, and equations may fail to render properly. Standard procedural text within the editor's letters has been deleted for the sake of brevity, but all official correspondence specific to the manuscript has been preserved.

Referees' reports, first round of review

Reviewer #1: Comments enter in this field will be shared with the author; your identity will remain anonymous. In this study, the authors investigate the involvement of airway epithelial cells in the genetic susceptibility to childhood-onset asthma (COA). By integrating transcriptomic data from epithelial cells infected with rhinovirus (RV) and genome-wide association studies (GWAS) data for asthma-related traits, the research offers insights into the mechanisms driving COA.

Comments

1. While exploring the role of airway epithelial cells in COA, the introduction should provide more comprehensive background information on the epidemiology and genetics of asthma, with a specific emphasis on childhood-onset asthma. Clearer explanations of the study's rationale, objectives, and main hypotheses would enhance the abstract's context.
2. The utilization of data from different time points after infection is noted in the second result. It is recommended that the authors explain the observed differences in heritability enrichment and significance between 24h and 42h, providing clarity on this temporal shift.
3. To enhance understanding, supplementary details and figures regarding the process of using CellphoneDB to identify potential ligand-receptor mechanisms involved in communication should be considered.
4. The second paragraph in the Results section introduces new hypotheses and results. Consideration should be given to either separating this paragraph into a distinct result with more detailed description or adjusting the title of the original result for a more comprehensive coverage.
5. More descriptive statistics, such as means and confidence intervals, or the use of tables and figures, would strengthen the presentation of quantitative results. For instance, providing quantitative summaries for genes at asthma risk loci upregulated with rhinovirus infection in airway epithelial cells would enhance clarity.
6. The discussion section should be expanded to include comparisons with previous studies (such as PMID:36778051, PMID:37285660), implications for asthma diagnosis and treatment, and future research directions. Addressing generalizability, reproducibility, potential sources of bias, and confounding factors will contribute to a more comprehensive evaluation.
7. The abstract is noted to be lengthy and detailed. A recommendation is made to condense the content, focusing on key findings and implications while avoiding abbreviations not defined in the abstract, such as RV.
8. Ensuring consistent formatting for subheadings and content in the Results section will enhance the overall structure and readability of the abstract.

Reviewer #2: The paper presents an interesting study investigating the genetic and environmental factors contributing to asthma, particularly focusing on childhood-onset asthma and its association with rhinovirus (RV) infection. The study integrates GWAS data with transcriptomic datasets of airway epithelial cells to explore how SNPs linked to different asthma phenotypes are expressed in different cell types, particularly airway epithelial cells, in response to RV infection.

One major point is that the adult asthma summary statistics contains around 40% of the number of GWS SNPs as childhood onset asthma, which may partially explain why there is so little enrichment of AOA associated genes. Nonetheless the COA findings look interesting but it is difficult to compare with AOA and also "All asthma" which is essentially unspecified asthma,

which may contain some CAO individuals. In addition there is a focus on T cells, but I note for example in COA, that 67% of the cells in Childhood-Onset Asthma with significant disease relevant score are Myeloid Cells, rather than T-cells, yet this does not appear to be discussed in the article, and may be an interesting finding.

Comments

Introduction:

Line 80 "In some children, rhinovirus wheezing does not lead to asthma". This sentence seems a little adrift at the end of the paragraph and may flow better earlier, perhaps as "Although it is clear that for some children rhinovirus wheezing does not lead to asthma, longitudinal epidemiological studies have shown that wheezing..."

Line 84: As a consequence, deciphering the mechanisms through which the risk alleles lead to disease is challenging; it has been achieved for only a small minority of risk loci. This should be referenced.

Line 113: unspecified-onset asthma is labelled as All-Asthma in the rest of the paper. That is a little confusing as this could imply Child-Onset + Adult-Onset asthma = All Asthma, rather than what it is, which is poorly annotated asthma. This could be better explained in the methods

Line 128: Could the authors provide further clarification why they used rheumatoid arthritis as a complex trait control for asthma.

Line 132: First, we sought to validate T-cell involvement in the genetic susceptibility to asthma. I am not sure I follow this. Have you not done work with T cells in order to validate the methods that you then subsequently use to demonstrate that airway epithelial cells are involved in the genetic susceptibility to asthma?

Figure S1 D vs E. In Figure S1 D you show the LDSC-SEG heritability enrichment coefficient for several Cell types, including pDCs. LDSC-SEG heritability enrichment coefficients are not presented for pDCs in Figure S1 E. If this is intentional, can you justify that in the methods as appropriate.

Line 138 to 142: You mention "In sinonasal mucosa tissue from healthy donors and chronic rhinosinusitis 141 patients, we observed that 21-83% of cells with significant disease relevant score (10% FDR) for 142 asthma-related traits are T cells (Supplementary Figure 2)" but there is no mention of the observation that 67% of the cells in Childhood-Onset Asthma with significant disease relevant score are Myeloid Cells. This appears to be quite striking, but does not appear to be discussed anywhere in the manuscript. Given that Myeloid cells represent innate leukocytes important in immune responses to viruses, perhaps discussing this observation may enhance the paper? Also this demonstrates differences between childhood asthma and all asthma.

Line 138-142 and Supplementary Figure 2) Could the authors speculate why there is no disease relevant score in Adult-onset asthma. In particular as there is "purple" cells in All Asthma, which may include Adult-onset asthma. Is Adult-Onset asthma "zero" here, could it reflect the low number of GWA AOA SNPs? Is it reasonable to say that "we observed that 21-83% of cells with significant disease relevant score (10% FDR) for asthma-related traits are T cells when this is zero for Adult-onset asthma?"

Lines 248 to 255: "For COA we identified 55 risk loci with genes upregulated upon rhinovirus infection in epithelial cells, 13 of which have L2G likely target genes (e.g. IL1RL1, IL4R, GSDMB, OVOL1, MYC), and 6 are the closest gene to the lead variant (e.g. IRF1, GPR183). For AOA only 19 risk loci have genes upregulated by RV in epithelial cells, among which 3 are likely target genes (e.g. IL4R, HDAC7, IL1RL1) and 3 are the closest gene to the lead variant (e.g. RAPGEF3, IRF1, SSR3)." Is it possible the smaller number of genes in AOA is reflected by the fact that the number of AOA GWS SNPs is ~40% of that in COA.

Lines 261-269:

You focus on MYC and OVOL1; it is not clear if these were the only genes that met a particular set of criteria for further analysis.

Lines 436 to 443

GWAS summary statistics, please state where you downloaded the preprocessed summary statistics from and also the accession numbers. I note that you used Ferreira et al for Childhood and Adult onset asthma summary statistics. There are 40 GWS SNPs in adult vs 98 in childhood. In addition there appears to be a higher proportion of SNPs with lower Odds Ratio's in AOA in comparison to COA. I note that you frequently do not see any enrichment in adult on set asthma (i.e. Fig 2c). Is there a possibility that this may be a function of the number of significant loci, And also a reflection that the effect sizes of many of those SNPs are lower than childhood? Especially as the more recent UKB "All Asthma" GWAS may contain some AOA individuals (but could be carried out on more recent genotyping chips?). The "All" asthma as well as the eczema GWAS was downloaded from ukbiobank, but as there is no accession number provided so is not possible to determine the volume of "asthma genes" in this trait that you use in your analysis. It is also unclear if this GWAS is published.

Authors' response to the first round of review

Reviewer #1: Comments enter in this field will be shared with the author; your identity will remain anonymous. In this study, the authors investigate the involvement of airway epithelial cells in the genetic susceptibility to childhood-onset asthma (COA). By integrating transcriptomic data from epithelial cells infected with rhinovirus (RV) and genome-wide association studies (GWAS) data for asthma-related traits, the research offers insights into the mechanisms driving COA.

RE: We thank the reviewer for the very helpful comments, which have allowed us to improve our manuscript. We have clarified the main text by adding more precise language throughout the manuscript and answered the reviewer's specific comments (see below). Furthermore, we have added new figures and tables to clarify our analyses and results.

Specific comments

1. While exploring the role of airway epithelial cells in COA, the introduction should provide more comprehensive background information on the epidemiology and genetics of asthma, with a specific emphasis on childhood-onset asthma. Clearer explanations of the study's rationale, objectives, and main hypotheses would enhance the abstract's context.

RE: We have added in the introduction more information about the epidemiology of asthma and specifically childhood-onset asthma:

"Although asthma can develop at any point in a person's life, it commonly begins in childhood. Longitudinal epidemiologic studies have demonstrated many risk factors for childhood-onset asthma. These include familial risk factors such as maternal and paternal asthma, maternal smoking and stress, perinatal risk factors such as preterm birth, low birth weight, C-section delivery, postnatal exposures including smoke, pollution, indoor allergens, and reduced microbiome diversity, and infections with respiratory syncytial virus and rhinovirus (Kuruvilla et al. 2019; Hui-Beckman et al. 2022). In the case of rhinovirus, wheezing with viral infection is a risk factor for developing asthma later in childhood (Holgate et al. 2015; Jackson and Gern 2022; Loxham and Davies 2017; Caliskan et al. 2013; Esquivel et al. 2017; Choi et al. 2021)."

We have also added additional information on the genetics of asthma:

"Asthma is significantly influenced by genetics, with heritability estimates starting at 35 % and going up to 95% (Ober & Yao, 2011). Genome-wide association studies (GWAS) have discovered more than 150 risk loci for asthma (Kim & Ober, 2019; Tsuo et al., 2022; Vicente et al., 2017). SNP-based heritability estimates in childhood-onset asthma (COA) are 2-3 times higher than that for adult-onset asthma (AOA) (M. A. Ferreira et al., 2019; Pividori et al., 2019). Accordingly, the number of discovered risk loci for COA is higher than for AOA (M. A. Ferreira et al., 2019; Pividori et al., 2019; Tsuo et al., 2022). Furthermore, the genetic correlation estimates between COA and AOA range from 0.63-0.78, indicating both shared and disease subset-specific factors (M. A. Ferreira et al., 2019; Tsuo et al., 2022). One of the most significant loci for COA is located at the 17q21-q12 locus, where ORM DL3 and GSDMB have been

Response to Reviewers

nominated as candidate causal genes (Kim and Ober 2019; Vicente, Revez, and Ferreira 2017). Notably, the association of 17q21-q12 variants with asthma is restricted to children who wheeze with rhinovirus, suggesting important gene

by environment interactions that are poorly understood (Calışkan et al. 2013).” Additionally, we have substantially modified the abstract. We removed several sentences to make it more concise, and we added some details to clarify our study’s rationale: “Asthma is a complex disease caused by genetic and environmental factors. Epidemiological studies have shown that wheezing during rhinovirus infection correlates with childhood asthma development. Genome-wide association studies (GWAS) have identified hundreds of non-coding genetic variants contributing to asthma susceptibility. Integrative analyses with transcriptomic and epigenomic datasets have indicated that T cells significantly contribute to asthma risk, which has been supported by mouse studies. However, these analyses lack data on airway epithelial cells, the first line of defense against respiratory viruses and allergens. Furthermore, large-scale transcriptomic T cell studies have not identified the regulatory effects of most asthma noncoding risk variants, indicating there could be additional cell types harboring these “missing regulatory effects”. We hypothesized that airway epithelial cells could mediate genetic susceptibility to asthma. Here we integrated GWAS data with transcriptomic datasets of airway epithelial cells subjected to different stimuli. We demonstrate that rhinovirus infection significantly upregulates childhood-onset asthma-associated genes, particularly in nonciliated cells. This enrichment is also observed with influenza infection but not with severe acute respiratory syndrome coronavirus 2 (SARS-CoV-2) or cytokine activation. Additionally, epithelial cells from patients with asthma showed a stronger heritability enrichment than cells from healthy individuals. Overall, our results suggest that rhinovirus infection is an environmental factor that interacts with genetic risk factors through non-ciliated airway epithelial cells to drive childhood-onset asthma.”

2. The utilization of data from different time points after infection is noted in the second result. It is recommended that the authors explain the observed differences in heritability enrichment and significance between 24h and 42h, providing clarity on this temporal shift. RE: We have compared the time-point-specific gene sets for 24 and 42 hours and have added the asthma-associated genes that were exclusive to the 24 hour set in Supplementary Table 4 for COA and Supplementary Table 5 for AOA. We have also added the following sentence in the results section to clarify the observed differences, along with Supplementary Figure 4D to show an example:

“While the lack of significance at 42 hours could be partly due to limited statistical power, we investigated which genes could be involved in the difference observed between 24 and 42 hours. We extracted the genes represented in the 24 hour-specific annotation but absent in the 42 hourspecific annotation and that were at GWAS risk loci (within 250 kb of the lead variant). We found 27 24 hour-exclusive GWAS genes for COA and 5 for AOA (Supplementary Tables 4-5). Among these is TSLP which encodes for a T2 cytokine, and has been associated with both asthma COA and AOA endotypes (Supplementary Figure 4D) (Sajuthi et al. 2022; Pividori et al. 2019; Lee et al. 2012; M. A. R. Ferreira et al. 2019).”

Supplementary Figure 4D

D

(D) Box plot depicting gene expression levels for TSLP in epithelial cells from the Basnet et al. dataset; boxes are colored by time point.

3. To enhance understanding, supplementardetails and figures regarding the process of using CellPhoneDB to identify potential ligand-receptor mechanisms involved in communication should be considered.

RE: We acknowledge that our report of the CellPhoneDB analysis was not complete. We have now expanded the description of our analysis and results in the main text, the methods section, and Supplementary Figure 6, including a cartoon that illustrates our analysis (Supplementary Figure 6A), the filtered results of CellPhoneDB (Supplementary Figure 6B), and the expression levels of the ligands and receptors in all cell subsets (Supplementary Figure 6C). In addition, we realized CellPhoneDB has been updated and now includes only curated pairs of interactors, hence we repeated our analysis using the new version 5, which we believe increases the confidence of the results. Consequently, we have lost the interactor pair we had originally mentioned in our manuscript. However, we now report all the candidate pairs found, and mention more than one example in the main text.

Revised main text:

"To investigate some of the possible ligand-receptor mechanisms through which cells may be communicating we used CellPhoneDB, which uses single cell RNA-seq data to identify expressed genes encoding ligands and receptors expressed by different cell clusters (Efremova et al. 2020). Specifically, we looked for ligand-encoding genes expressed in RV-infected ciliated cells and their corresponding receptor-encoding gene expressed in non-infected nonciliated cells, with the rationale that the receptor should already be expressed in the non-infected state, ready to receive the signal by the ligand produced by ciliated cells upon direct infection by the virus (Supplementary Figure 6A). Furthermore, we required that the ciliated cell ligand-encoding gene is significantly upregulated upon rhinovirus infection (5% FDR, \log_2 fold change > 0.5). We identified eleven candidate pairs of interactors (Supplementary Figure 6B-C). Among these ligand-receptor pairs are TNFSF10-TNFRSF10B. TNFSF10 encodes for the TRAIL cytokine, which belongs to the TNF ligand family, and it serves as ligand to the receptor encoded by TNFRSF10B (a.k.a.

DR5, TRAILR2 or CD262), which can induce apoptosis. A study using a rhinovirus mouse model and in vitro activation of a human airway epithelial cell line suggested that TRAIL promotes RV-induced airway hyperreactivity and inflammation (Girkin et al., 2017). Another two pairs of ligand-receptor interactors that we identified involve the Epidermal Growth Factor Receptor (EGFR), which has been reported to be induced by rhinovirus and other respiratory viruses and utilized for suppressing interferon signaling and inducing higher viral replication (Kalinowski et al., 2014; Ueki et al., 2013)."

Revised methods:

"To identify potential pairs of interactors between RV-infected ciliated and non-infected non-ciliated epithelial cells we used the dataset of Basnet et al. 2023. We first retrieved two subsets of the cells contained in this dataset, specifically: RV-infected ciliated cells and non-infected non-ciliated cells. We identified hypothetical pairs of interactors (ligand-receptor) between RVinfected ciliated cells and non-infected non-ciliated cells with CellPhoneDB (v5) with the method "statistical_analysis" (default parameters). We then filtered the results obtained by CellPhoneDB, by retrieving significant ligand-receptor interactions (permutation $P < 0.05$). We also requested that the ligand be upregulated upon rhinovirus infection within ciliated cells. To evaluate which genes were upregulated between rhinovirus infected ciliated cells and noninfected non-ciliated cells, we used the Seurat function "FindMarkers" (Wilcoxon test, min.pct=0.1 and default parameters). We selected upregulated genes based on $\text{Log}_2\text{FC} > 0.50$ and $P \text{ adjusted} < 0.05$."

Revised Supplementary Figure 6:

Supplementary Figure 6

4. The second paragraph in the Results section introduces new hypotheses and results. Consideration should be given to either separating this paragraph into a distinct result with more detailed description or adjusting the title of the original result for a more comprehensive coverage.

RE: We have changed the subheadings in the results section to improve clarity and consistency in style across them. We paste the subheadings below:
 "T cell involvement in asthma is confirmed by methods integrating cell-typespecific profiles and GWAS data.
 Infection with rhinovirus A16 induces upregulation of asthma-associated genes in epithelial cells from healthy donors.
 Infection with rhinovirus C15 induces upregulation of asthma-associated genes in epithelial cells from healthy donors.
 Rhinovirus-induced asthma-associated genes are specifically enriched in nonciliated epithelial cells.
 Enrichment of asthma-associated genes after rhinovirus infection is stronger

in epithelial cells from asthma patients.

Likely target genes at asthma risk loci are upregulated in airway epithelial cells after rhinovirus infection.

Not all viral infections significantly upregulate asthma-associated genes in epithelial cells.

Stimulation with type 2-related cytokines does not induce significant upregulation of asthma-associated genes in epithelial cells."

5. More descriptive statistics, such as means and confidence intervals, or the use of tables and figures, would strengthen the presentation of quantitative results. For instance, providing quantitative summaries for genes at asthma risk loci upregulated with rhinovirus infection in airway epithelial cells would enhance clarity.

RE: We have added summaries as supplementary tables for genes at asthma risk loci that are upregulated in epithelial cells upon rhinovirus infection at 24 hours in either Helling et al. or Basnet et al. (Supplementary Tables 4-5). Additionally, we have added tables of the genes and associated differential expression summary statistics that ranked in the top 10% by tstatistic in each of the analyses throughout the manuscript (Supplementary Table 3a-n).

Furthermore, we added the raw LDSC-SEG results as well as the normalized coefficient τ^* for each of the heritability enrichment analyses (Supplementary Table 2a-j).

6. The discussion section should be expanded to include comparisons with previous studies (such as PMID:36778051, PMID:37285660), implications for asthma diagnosis and treatment, and future research directions. Addressing generalizability, reproducibility, potential sources of bias, and confounding factors will contribute to a more comprehensive evaluation.

RE: We thank the reviewer for pointing us to these studies. We have added a paragraph in the discussion comparing the Bao et al. study to ours and potential implications for asthma treatment:

"Our study highlighted genes in GWAS loci that get upregulated with RV in airway epithelial cells. Twelve of these are within the list of leading prioritized genes as potential drug targets reported by a recent study integrating asthma GWAS information with protein-protein interaction data (IL-6, MYC, PRKCQ, ETS1, IL-4R, IRF1, IL-1R2, RELA, CDK2, SOCS1, NFKB2, PSMA6) (Bao et al. 2023).

Of these, IL-6 is a target for a therapy that is in Phase II clinical trials to treat severe asthma (Clazakizumab), and IL-4R is a target for Dupilumab, which has been tested in clinical trials for uncontrolled severe asthmatics (Ricciardolo, Bertolini, and Carriero 2021). Our study underscores the possibility that these therapies might in part act through airway epithelial cells and raises the consideration of future therapies being designed as to specifically deliver to airway epithelial cells and/or when genetically susceptible subjects are infected with rhinovirus. In fact, GSK has a Phase I clinical trial on an inhalation powder for treatment during rhinovirus infection in subjects with mild asthma ("Efficacy and Safety of GSK3923868 Inhalation Powder, During Experimental Human Rhinovirus Infection in Participants With Mild Asthma," n.d.)."

We have added findings from the Tsuo et al. study in the introduction, as mentioned above in response to Point 1. In addition, we have cited Tsuo et al. and an additional study in the discussion, while expanding the limitations of our study and future directions:

"Another limitation of our study is that it primarily focuses on GWAS data derived from European ancestry individuals, partly due to the bias in GWAS studies so far (Martin et al, 2019), and consequently limitations posed by the tools utilized in our study, which predominantly leverage data from specific populations. A recent multi-ancestry meta-analysis for asthma highlighted that genetic effects are largely consistent between ancestries, but that ascertaining all ancestries is important to find all of the risk loci for asthma, as there are risk alleles that are frequent in some populations but not others (Tsuo et al. 2022). Broadening the scope of heritability enrichment analyses to incorporate multi-ancestry meta-analyses would help to further characterize the cell types and cell states relevant for asthma and its related traits. Moreover, given that the cases in the GWAS utilized in this study are based on self-reported doctor-diagnosed disease and PheCodes from the UK Biobank, inaccuracies in patient classification are expected, and replication of findings using summary statistics from future GWAS involving better characterized patients and controls is important."

7. The abstract is noted to be lengthy and detailed. A recommendation is made to condense the content, focusing on key findings and implications while avoiding abbreviations not

defined in the abstract, such as RV.

RE: We have reduced the length of the abstract concentrating on the essential findings and we have removed abbreviations that were not defined. The new abstract is pasted above in response to Point 1.

8. Ensuring consistent formatting for subheadings and content in the Results section will enhance the overall structure and readability of the abstract.

RE: We have reviewed the subheadings in the results section as mentioned in Point 4, maintaining a consistent style.

Reviewer #2: The paper presents an interesting study investigating the genetic and environmental factors contributing to asthma, particularly focusing on childhood-onset asthma and its association with rhinovirus (RV) infection. The study integrates GWAS data with transcriptomic datasets of airway epithelial cells to explore how SNPs linked to different asthma phenotypes are expressed in different cell types, particularly airway epithelial cells, in response to RV infection. One major point is that the adult asthma summary statistics contains around 40% of the number of GWS SNPs as childhood onset asthma, which may partially explain why there is so little enrichment of AOA associated genes. Nonetheless the COA findings look interesting but it is difficult to compare with AOA and also "All asthma" which is essentially unspecified asthma, which may contain some CAO individuals. In addition there is a focus on T cells, but I note for example in COA, that 67% of the cells in Childhood-Onset Asthma with significant disease relevant score are Myeloid Cells, rather than T-cells, yet this does not appear to be discussed in the article, and may be an interesting finding.

RE: We thank the reviewer for raising up these points. We have addressed the reviewer's comments, resulting in significant improvements of our manuscript. We provide answers to the specific comments below.

Specific comments

Introduction:

Line 80 "In some children, rhinovirus wheezing does not lead to asthma". This sentence seems a little adrift at the end of the paragraph and may flow better earlier, perhaps as "Although it is clear that for some children rhinovirus wheezing does not lead to asthma, longitudinal epidemiological studies have shown that wheezing..."

RE: We agree this sentence was adrift. We have removed it and performed modifications to that paragraph (to also include suggestions from Reviewer 1). That paragraph now reads as follows:

"Asthma is a complex and heterogeneous disease that affects 300 million children and adults worldwide and represents a significant burden to healthcare (\$82 billion for the US in 2013)(Porsbjerg et al. 2023). Although asthma can develop at any point in a person's life, it commonly begins in childhood. Longitudinal epidemiologic studies have demonstrated many risk factors for childhood-onset asthma. These include familial risk factors such as maternal and paternal asthma, maternal smoking and stress, perinatal risk factors such as preterm birth, low birth weight, C-section delivery, postnatal exposures including smoke, pollution, indoor allergens, and reduced microbiome diversity, and infections with respiratory syncytial virus and rhinovirus. In the case of rhinovirus, wheezing with viral infection is a risk factor for developing asthma later in childhood (Jackson and Gern 2022; Loxham and Davies 2017; Holgate et al. 2015). These observations have led to two hypotheses: (1) rhinovirus infection could be causal in asthma development or (2) rhinovirus-induced wheeze is a biomarker that identifies children at increased risk for asthma development."

Line 84: As a consequence, deciphering the mechanisms through which the risk alleles lead to disease is challenging; it has been achieved for only a small minority of risk loci. This should be referenced.

RE: After reading several studies that provide insights into possible disease mechanisms driven by asthma risk alleles, we have realized it is hard to claim that the mechanisms have indeed been totally deciphered in these loci. Hence, to avoid the risk of over-claiming, we have removed the last part of the sentence. It now reads as follows:

"As a consequence, deciphering the mechanisms through which the risk alleles lead to disease is challenging."

Line 113: unspecified-onset asthma is labelled as All-Asthma in the rest of the paper. That is a little confusing as this could imply Child-Onset + Adult-Onset asthma = All Asthma, rather than what it is, which is poorly annotated asthma. This could be better explained in the methods

RE: We agree with the reviewer. We have changed our label of "All Asthma" to "Unspecified-Onset Asthma" (UOA) and we now provide more details in the methods section. "The adult-onset and childhood-onset asthma are GWAS with careful curation of the UK Biobank data and therefore include fewer individuals (case/control numbers in Supplementary Table 1, for details on patient selection criteria please see Ferreira et al.(M. A. R. Ferreira et al. 2019)). The adult-onset asthma GWAS was performed with patients with age at first diagnosis between 20 and 60 years of age. The childhood-onset asthma GWAS was performed with patients with age at first diagnosis of 19 years of age or younger. The unspecified onset-asthma GWAS was performed with less meticulous curation of UK Biobank data and includes individuals with any age of onset (Sudlow et al., 2015)."

Line 128: Could the authors provide further clarification why they used rheumatoid arthritis as a complex trait control for asthma.

RE: We have added a sentence in the results section to justify why we used rheumatoid arthritis as a complex trait control for asthma.

"Throughout our analyses we included three complex traits as controls: height, as a non-immune control, Alzheimer's disease (AD) as a trait implicating myeloid cells, and rheumatoid arthritis (RA) as a lymphocyte-driven disease with a strong T cell component (Zhang et al. 2022; Finucane et al. 2018; Soskic et al. 2019; Calderon et al. 2019; Trynka et al. 2013). For both asthma and RA, activated T cells have been shown to be a relevant cell state (Calderon et al., 2019; Soskic et al., 2022), however, RA is an autoimmune disease that affects the joints, therefore, a genetic enrichment for RA in gene expression of airway epithelial cell states is not expected."

Line 132: First, we sought to validate T-cell involvement in the genetic susceptibility to asthma. I am not sure I follow this. Have you not done work with T cells in order to validate the methods that you then subsequently use to demonstrate that airway epithelial cells are involved in the genetic susceptibility to asthma?

RE: We have clarified our sentences to explain better our goals:

"First, we sought to confirm the effectiveness of the methods to identify T cells as relevant for the genetic susceptibility to asthma, as previously reported in the literature (Finucane et al. 2018; Calderon et al. 2019; Zhang et al. 2022; Soskic et al. 2019), and to test the methods' resolution to pinpoint the specific T cell states and subsets driving the signal. "

Figure S1 D vs E. In Figure S1 D you show the LDSC-SEG heritability enrichment coefficient for several Cell types, including pDCs. LDSC-SEG heritability enrichment coefficients are not presented for pDCs in Figure S1 E. If this is intentional, can you justify that in the methods as appropriate.

RE: Yes, this was intentional because there was no stimulation condition for pDCs and plasma cells, so we could not include them in the specific analysis which tests differential accessibility between the resting and activated state. We have added the following in the legend to clarify: "For plasma cells and pDCs there was no in vitro activation performed in the original study. Cell-type-specific results shown for these cell types in panel D depict differential accessible peaks in resting state only."

Line 138 to 142: You mention "In sinonasal mucosa tissue from healthy donors and chronic rhinosinusitis 141 patients, we observed that 21-83% of cells with significant disease relevant score (10% FDR) for 142 asthma-related traits are T cells (Supplementary Figure 2)" but there is no mention of the observation that 67% of the cells in Childhood-Onset Asthma with is significant disease relevant score are Myeloid Cells. This appears to be quite striking, but does not appear to be discussed anywhere in the manuscript. Given that Myeloid cells represent innate leukocytes important in immune responses to viruses, perhaps discussing this observation may enhance the paper? Also this demonstrates differences between childhood asthma and all asthma.

RE: We thank the reviewer for pointing this out, and we agree that this finding could be of relevance. We have added a sentence explicitly describing this finding in the results section: "In addition, we found that 67% of cells for childhood-onset asthma are myeloid cells (10% FDR)."

We have added the following to the discussion:

"Another cell type that showed significant over-expression of genes in COA risk loci was the myeloid lineage in nasal samples from chronic rhinosinusitis patients and healthy controls (Wang et al. 2022) (Supplementary Fig. 2). This

is in line with the study of macrophages in the context of asthma (Britt et al. 2023; van der Veen, de Groot, and Melgert 2020) and with the findings of a recent large scale single cell RNA-seq study in lung where they reported colocalization between COA risk loci and expression quantitative trait loci in macrophages, monocytes, and dendritic cells (Natri et al. 2023). Together, these findings suggest myeloid cells from the respiratory tract could be another important cell type worth studying further to find new genetic mechanisms for COA."

Line 138-142 and Supplementary Figure 2) Could the authors speculate why there is no disease relevant score in Adult-onset asthma. In particular as there is "purple" cells in All Asthma, which may include Adult-onset asthma. Is Adult-Onset asthma "zero" here, could it reflect the low number of GWA AOA SNPs? Is it reasonable to say that "we observed that 21-83% of cells with significant disease relevant score (10% FDR) for asthma-related traits are T cells when this is zero for Adult-onset asthma?"

RE: We thank the reviewer for pointing this out, which can be relevant for the interpretation of results in the manuscript. As highlighted by the reviewer, AOA has fewer genome-wide significant risk loci, despite having a larger sample size in the GWAS. In addition, for shared risk loci between AOA and COA, AOA presents lower odds ratio values. Furthermore, SNPbased heritability estimates are 2-3 times larger for COA than for AOA. Regarding the analysis pointed out by the reviewer, at a more relaxed FDR threshold (20%), we observe that 73% of the significant scDRS cells for AOA are T cells. However, a stronger genetic component for COA than for AOA cannot fully explain the lack of significant scDRS cells at 10% FDR for AOA in the Wang et al. dataset of nasal samples. In the dataset of peripheral T cells stimulated with dust mite (Seumois et al., 2020), we found no significant scDRS cells for COA, but we did find significant cells for AOA. We think perhaps in this scenario, it could mean that the T cell signal for COA is more generalized and not particular for specific T cell subsets, given this dataset has only T cells and the control genes in scDRS are taken from the same T cells. Conversely, for the negative result for AOA in the Wang et al. dataset, we could speculate that perhaps the T cells in those samples (coming from sinonasal mucosa tissue from 5 healthy donors and 16 CRS patients) were not in a high enough relevant cell state relative to the other cell types and states in the dataset to show significant scDRS cells.

We would also like to highlight that the way the scDRS works is by comparing the disease score with the control scores based on 1000 control gene sets that are matched by gene expression levels and variance to the disease genes. In the disease score, genes are weighted based on the MAGMA z-scores. In our study, MAGMA z-scores are, as expected, on average higher in COA (3.27) than in AOA (2.68). However, in scDRS the MAGMA z-scores are also used as weights for calculating the control scores (using the matching disease gene weight). Hence, the over-expression of disease associated genes is assessed relative to very wellmatched control gene sets taken from the same dataset. Hence, we wouldn't expect that a higher heritability or a higher number of genome-wide significant loci would impact the disease relevant score statistical significance. In fact, UOA has even higher MAGMA z-scores (mean = 4.04) than COA, and despite this, COA had over 5-fold more significant cells than UOA in the single cell RNA-seq dataset with RV-infected airway epithelial cells (Fig. 2). This goes in line with the results by LDSC-SEG in the RV datasets, in which COA had larger heritability enrichment coefficients than UOA (and AOA). LDSC-seg uses most SNPs tested in a GWAS (not only those that pass the genome-wide significance threshold) and assesses heritability enrichment relative to each disease heritability, so we also do not expect to have strong biases dependent on the number of genome-wide significant loci for these polygenic diseases using this method. In fact, for the influenza upregulated genes, we observed larger heritability enrichment coefficient in AOA compared to COA and UOA (Supplementary Fig. 9). We have added some sentences in the introduction to make the reader more aware about the genetic differences between COA and AOA from the beginning:

"SNP-based heritability estimates in childhood-onset asthma (COA) are 2-3 times higher than that for adult-onset asthma (AOA) (Pividori et al. 2019; M. A. Ferreira et al. 2019). Accordingly, the number of discovered risk loci for COA is higher than for AOA (M. A. Ferreira et al. 2019; Pividori et al. 2019; Tsuo et al. 2022). Furthermore, the genetic correlation estimates between COA and AOA range from 0.63-0.78, indicating both shared and disease subset-specific factors (M. A. Ferreira et al. 2019; Tsuo et al. 2022)."

Additionally, we have clarified the statement pointed out by the reviewer in the results section with an additional sentence:

"In sinonasal mucosa tissue from healthy donors and chronic rhinosinusitis

patients, we observed that 21-83% of cells with significant disease relevant score (10% FDR) for asthma-related traits are T cells (Supplementary Figure 2). AOA did not have any cells with significant disease relevant score at 10% FDR, however, at 20% FDR 294 out of 405 cells (73%) are T cells."

As a side note, for the case of COA in the house dust mite T cell dataset, we found that COA did not have any cells with significant disease relevant score at 10% FDR (nor at 20%), however at 21% FDR we identified 9 T effector and 7 Th2 cells that were significant. Due to the unusual FDR threshold and the very few numbers of significant hits for COA we decided not to report this result in the main manuscript.

Finally, in the discussion, we introduced a few new sentences that highlight the lower number of genome-wide significant loci for AOA compared to COA, and how this may influence the interpretation of results. Below we paste this fragment of the discussion:

"Here, we observe that in non-ciliated airway epithelial cells rhinovirus induces upregulation of GSDMB as well as putative causal genes in 54 additional risk loci. This demonstrates a widespread interaction between in vitro rhinovirus infection and polygenic susceptibility to childhood-onset asthma, specifically mediated through airway epithelial cells. In contrast, 19 risk loci for adult-onset asthma have genes that get upregulated with rhinovirus infection in airway epithelial cells. This result is in part due to the lower number of genome-wide significant loci in AOA (56) compared to COA (126), even if the AOA GWAS had a larger sample size. In fact, AOA presented positive heritability enrichment coefficients for RV-upregulated genes in airway epithelial cells (Fig. 1 and Fig. 3), however, for most analyses, the heritability enrichment was not statistically significant. This contrasts with the T cell specific annotations, for which AOA had statistically significant enrichments in most analyses. Hence, our data suggests that RV-infected airway epithelial cells may play a more important role in mediating genetic susceptibility to COA compared to AOA. These findings are concordant with a previous study reporting that genes at COA-specific risk loci (as compared to AOA) have high expression in skin, which is a barrier tissue with an abundance of epithelial cells (Pividori et al. 2019)."

Lines 248 to 255: "For COA we identified 55 risk loci with genes upregulated upon rhinovirus infection in epithelial cells, 13 of which have L2G likely target genes (e.g. IL1RL1, IL4R, GSDMB, OVOL1, MYC), and 6 are the closest gene to the lead variant (e.g. IRF1, GPR183). For AOA only 19 risk loci have genes upregulated by RV in epithelial cells, among which 3 are likely target genes (e.g. IL4R, HDAC7, IL1RL1) and 3 are the closest gene to the lead variant (e.g. RAPGEF3, IRF1, SSR3)." Is it possible the smaller number of genes in AOA is reflected by the fact that the number of AOA GWS SNPs is ~40% of that in COA.

RE: We agree with the reviewer this could be due to a larger number of genome-wide significant loci for COA, although potentially also due to possibly stronger RV-activated epithelial cell involvement for COA than for AOA. To avoid biasing the reporting of results, we have removed the word "only" when reporting the AOA results, and as mentioned above, we have added details of the genetic differences between AOA and COA in the introduction, and we have added some sentences discussing this point in the discussion.

Lines 261-269: You focus on MYC and OVOL1; it is not clear if these were the only genes that met a particular set of criteria for further analysis.

RE: We have added the following sentences and Supplementary Tables 6-9 to clarify our selection criteria and report the other genes that passed the criteria:

"In this T cell dataset, we identified genes that were upregulated at 24 hours post-stimulation when compared to resting at 5% FDR."

"Other RV-induced asthma-associated genes in epithelial cells that are also upregulated in activated T cells are reported in Supplementary Tables 6-7, and those which are not significantly up-regulated in activated T cells are reported in Supplementary Tables 8-9."

Lines 436 to 443

GWAS summary statistics, please state where you downloaded the preprocessed summary statistics from and also the accession numbers. I note that you used Ferreira et al for Childhood and Adult onset asthma summary statistics. There are 40 GWS SNPs in adult vs 98 in childhood. In addition there appears to be a higher proportion of SNPs with lower Odds Ratio's in AOA in comparison to COA. I note that you frequently do not see any enrichment in adult on set asthma (i.e. Fig 2c). Is there a possibility that this may be a function of the number of significant loci, And also a reflection that the effect sizes of many of those SNPs

are lower than childhood? Especially as the more recent UKB "All Asthma" GWAS may contain some AOA individuals (but could be carried out on more recent genotyping chips?). The "All" asthma as well as the eczema GWAS was downloaded from ukbiobank, but as there is no accession number provided so is not possible to determine the volume of "asthma genes" in this trait that you use in your analysis. It is also unclear if this GWAS is published.

RE: We have added in the methods section the link to where we downloaded our preprocessed summary statistics:

"We downloaded the pre-processed summary statistics from Alkes Price website: <https://console.cloud.google.com/storage/browser/broad-alkesgrouppublic-requester-pays/UKBB>."

For the miami plot shown in Figure, 4 we downloaded the raw summary statistics from the GWAS catalog and we have added the accession numbers:

"For visualization of GWAS SNPs in Figure 4, we downloaded the childhoodonset asthma and adult-onset asthma summary statistics from the GWAS catalog (GCST007800,GCST007799)."

Links to all GWASs and omic studies used are now also available in the STAR Methods Key Resources Table.

In addition, as mentioned above in response to other points of the reviewer: we have now added in the introduction several facts depicting the differences in AOA and COA genetics, we have added additional details in the methods about the patient criteria used for the AOA, COA, and UOA GWASs, with their corresponding references, we have added Supplementary Table 1 with number of cases and controls for each GWAS used in this study, and we have expanded our discussion to highlight the point of AOA having less number of genome-wide significant loci.

Referees' reports, second round of review

Reviewer #1: Authors have extensively revised the manuscript, recommended for publication.

Reviewer #2: The authors have answered my comments

Authors' response to the second round of review

Point-by-point responses to review comments (Our responses are in blue).

Reviewer #1

Comments: I would like to thank the authors for putting a lot of effort in revising the manuscript. While it is in a good shape now, the paper still suffers from very complicated figure panels. I can follow the logical flow now (but I have also now read it 5-6 times). I worry that the way it is presented with many many figure panels is off putting and readers will find it difficult to follow the logic. I would insist that the authors make more of an effort in thinking about what panels are actually needed in the main figures and modify the manuscript further.

Response: Thank you for your detailed comments and for acknowledging our efforts in revising the manuscript. We understand your concerns regarding the complexity of the figure panels. In response, we have streamlined the figures by moving some of the detailed results from Figures 2, 5, and 6 to the supplementary materials. This adjustment has allowed us to present a more concise and clear representation of our findings in the main figures. We believe this will enhance the readability and logical flow of the manuscript. We appreciate your patience and constructive suggestions that have significantly improved our work.

Reviewer #2

Comments: All my suggestions are addressed. No further comments.

Response: Thank you for your positive feedback and for confirming that all your suggestions have been addressed. We appreciate your thorough review and valuable comments that have contributed to strengthening our manuscript.